# Central dogma rates and the trade-off between precision and economy in gene expression

Jean Hausser [1], Avi Mayo[1], Leeat Keren[1] & Uri Alon[1]

Steady-state protein abundance is set by four rates: transcription, translation, mRNA decay and protein decay. A given protein abundance can be obtained from infinitely many combinations of these rates. This raises the question of whether the natural rates for each gene result from historical accidents, or are there rules that give certain combinations a selective advantage? We address this question using high-throughput measurements in rapidly growing cells from diverse organisms to find that about half of the rate combinations do not exist: genes that combine high transcription with low translation are strongly depleted. This depletion is due to a trade-off between precision and economy: high transcription decreases stochastic fluctuations but increases transcription costs. Our theory quantitatively explains which rate combinations are missing, and predicts the curvature of the fitness function for each gene. It may guide the design of gene circuits with desired expression levels and noise.

[1] Department of Molecular Cell Biology, Weizmann Institute of Science, Rehovot 76100, Israel. Correspondence and requests for materials should be addressed to J.H. (email: jean@hausser.org) or to U.A. (email: uri.alon@weizmann.ac.il)

To function well in a given environment, cells need to express genes at the right protein copy number[1–3]. Steady-state protein abundance is set by two reactions of synthesis —transcription and translation—balanced by two processes of decay—dilution and degradation of mRNAs and proteins[4]. Together, these make up the four basic rates of the central dogma[5].

The rates of these four central dogma reactions are controlled by diverse regulators. Transcription rate is set by transcription factors and chromatin remodelers[6]. Translation is modulated by RNA-binding proteins and non-coding RNAs[7,8], and so on. The effects of these molecular controls can be summarized by the central dogma rates, such that each protein is a point in a four dimensional space whose axes are the four rates. In this study, we name this the Crick space, in honor of Francis Crick who proposed the central dogma[5].

One important property of Crick space is that the same steady-state protein abundance can be achieved by many combinations of rates. For example, consider a protein made at 1000 copies per hour (Fig. 1). This can be achieved by transcribing 100 mRNAs and translating 10 proteins from each mRNA every hour. Alternatively, the 1000 proteins could be made from one mRNA translated into 1000 proteins per hour (in this example, we fixed mRNA and protein decay). There is an infinite number of ways to combine transcription and translation rates, $\beta_m$ [mRNA h$^{-1}$] and $\beta_p$ [protein mRNA$^{-1}$ h$^{-1}$], in order to supply a given steady-state number of proteins $p$, namely $\beta_m\beta_p/\alpha_m\alpha_p = p$ where $\alpha_m$ [h$^{-1}$] and $\alpha_p$ [h$^{-1}$] are the rates of mRNA and protein decay by dilution and degradation (Methods).

Here we ask whether such combinations occur randomly, as expected if they are equally beneficial and historical accident or genetic drift is at play, or whether there are rules based on specific translation/transcription ratios that have selective advantage. If such rules exist, we might expect to see patterns in the way that genes occupy Crick space.

While there has not been systematic evidence for rules so far, previous work described how different combinations of central dogma rates can differ in their biological impact. One line of work shows that intrinsic noise[9–12], the stochastic variation in protein number due to small-number effects, is largest when there are few mRNAs translated into many proteins[13–16]. This large noise occurs because the relative fluctuations in the number of a few mRNAs are large, and are amplified by strong translation. This idea was first proposed by McAdams & Arkin[13] based on theoretical arguments. The prediction that a given protein abundance can be reached with the least noise when transcription is high and translation is low was validated by Ozbudak et al.[14] using synthetic constructs with defined transcription and translation rates. Bar-Even et al.[16] measured the noise and abundance of 43 *S. cerevisiae* proteins and found that noise scaled with protein abundance in a way consistent with the predictions of McAdams & Arkin[13]. Also in *S. cerevisiae*, Newman et al.[17] observed a correlation between noise and mRNA abundance for 2500+ genes.

Another difference between combinations of rates that give the same steady-state protein abundance is mRNA cost—the reduction in fitness due to production of mRNA[18–22]. Theoretical studies proposed that the cost of synthesizing mRNAs confers a selectable disadvantage[13,18,19]. Experiments in *S. cerevisiae*[21] indicate that expressing a non-beneficial mRNA penalizes the growth rate in proportion to the transcription rate (Methods). In *E. coli*, expressing a protein at a given abundance from a larger number of mRNAs decreases fitness[22].

Noise and cost are thought to be significant components in determining the fitness and selection of biological designs[13,23–27]. In particular McAdams & Arkin[13], and others[14] proposed that

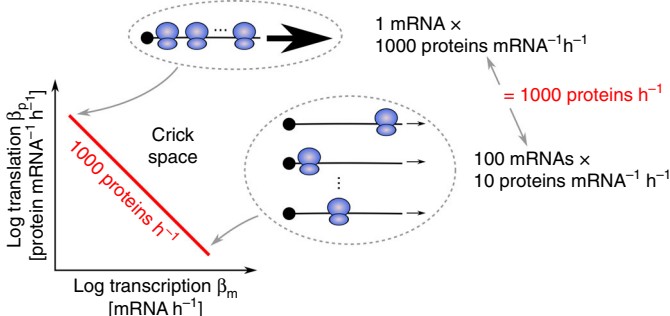

**Fig. 1** For each gene, an infinite number of combinations of transcription and translation rates can achieve a given protein abundance. For example, to obtain 1000 proteins per hour, one possibility is to translate 1000 proteins per hour from a single mRNA. Another option is to translate 10 proteins per hour from 100 mRNAs. We assumed fixed mRNA and protein decay to simplify the visualization. See also Supplementary Fig. 1

there should be a trade-off between minimizing gene expression costs and minimizing noise in protein abundance. Testing this hypothesis has been difficult, partly because the central dogma rates could not be measured genome-wide until recently[28]. Another hurdle that prevented testing the hypothesis of a precision-economy trade-off in gene expression is that it is unclear how the interplay of precision and economy should affect the distribution of genes in the Crick space.

Here we address the question of rules for protein expression by analyzing comprehensive data on central dogma rates from several model organisms[3,17,26,29–31] and by theory on evolutionary trade-offs. We find that about half of the Crick space is empty: genes do not seem to combine high transcription with low translation. This depleted region is accessible by synthetic constructs, and hence its emptiness is not based on mechanistic constraints. We explain the empty Crick space by a trade-off between cost and noise of gene expression. This theory accurately predicts the boundary of the empty region which varies by 2 orders of magnitude between the model organisms we considered. This approach might be of use to design synthetic gene expression circuits, and suggests rules for central dogma rates that seem to apply from bacteria to humans.

## Results

**Genes combining high transcription and low translation are depleted.** We estimated transcription $\beta_m$ and translation $\beta_p$ rates, as well as mRNA and protein decay rates, for thousands of genes from previous mRNAseq and ribosome profiling (RP) experiments in *S. cerevisiae*[29], *M. musculus*, *H. sapiens*[30], and *E. coli*[3] (Methods). All data were collected under conditions of rapid growth.

We find, in accordance with previous studies[3,26,32,33], that transcription and translation rates in rapidly growing cells vary much more from gene to gene than mRNA and protein decay rates. Transcription and translation rates vary over a 1000-fold range compared to a 10-fold range for decay rates of mRNA and protein (Supplementary Fig. 1a–d). Taking into account gene-specific mRNA and protein decay rates has only a small impact on the position of genes in 2D Crick space (Supplementary Fig. 1e–g, Methods). We therefore simplify our discussion by considering a 2D Crick space, formed by transcription and translation rates.

Reducing the 4-dimensional Crick space to two dimensions neglects aspects of cell biology such as the dynamics of gene regulation in response to environmental perturbations[34,35], but it allows us to focus on the most variable rates in setting steady-state protein abundance in growing cells, transcription and

translation, and ask what rules may underlie them. Furthermore, reducing to two dimensions yields a more complete picture of Crick space (mRNA and protein decay rates have typically been measured for 20–50% of genes) and avoids the concern that the decay rates have been measured in separate studies, unlike the synthesis rates for *S. cerevisiae*, *E. coli*, and *H. sapiens* (Methods).

Plotting the transcription and translation rates of genes in four model organisms, we observe common boundaries in the Crick space (Fig. 2). First, the maximal translation is $10^{3.6}$–$10^4$ proteins per mRNA per hour, a bound that can be explained from the ribosome translocation speed (Methods). Second, we observed lower bounds on the transcription rate and on the product of transcription and translation. These boundaries stem from technical limits of the assays (Supplementary Fig. 2a).

Unexpectedly, and most importantly for the present study, there was a lack of genes combining high transcription with low

translation (blue regions in Fig. 2). We call this region the depleted region of the Crick space. This depleted region makes up about half of the Crick space and is bounded by a line of constant ratio between transcription $\beta_m$ and translation $\beta_p$, namely $\beta_p/\beta_m = k$, with $k = 1.1 \pm 0.1$, $14 \pm 3$, $44 \pm 3$, and $66 \pm 4$ in *S. cerevisiae*, *E. coli*, *M. musculus*, and *H. sapiens*, respectively (± represents standard error, Table 1). In logarithmic axes, the boundary of the depleted region has slope 1 and intercept $\log(k)$. Hence, $k$ determines the boundary of the depleted region.

In *E. coli*, the boundary of the depleted region has additional structure. Departing from the main distribution of genes are a set of 59 genes with very high transcription and translation rates ($\beta_m > 80 \, h^{-1}$ and $\beta_p > 1000 \, mRNA^{-1}h^{-1}$, indicated by an arrow in Fig. 2d). About 90% of these genes are ribosomal proteins. The rest are high abundance proteins such as the glycolysis enzyme *gapA* (the *E. coli* equivalent of *Gapdh*), the ATP synthase c subunit, and outer-membrane proteins (OMPs). Most genes in

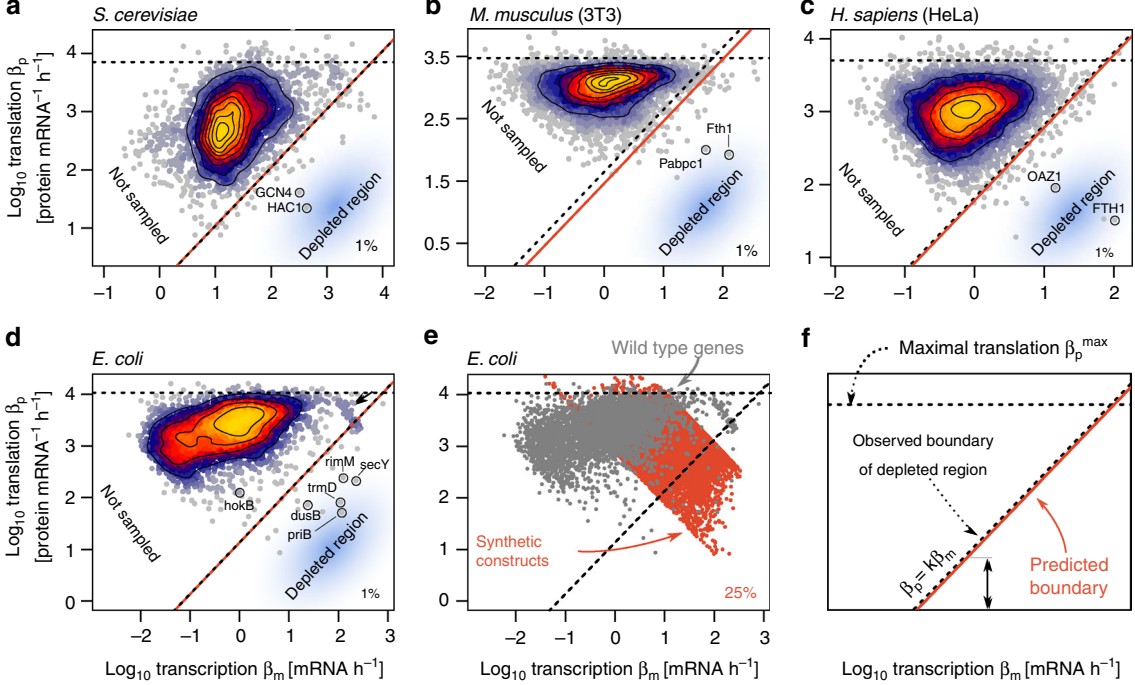

**Fig. 2** Genes combining high transcription and low translation are depleted from the Crick space across organisms. **a–d** There is a depleted region in the Crick space of four model organisms. Transcription and translation rates were estimated from ribosome profiling and mRNA sequencing data. The top percentile of translation rates $\left(\beta_p^{max}\right)$ is represented as a horizontal dashed line. The observed boundary of the depleted region (diagonal dashed line) has slope 1 and is such that 99% of the genes have a larger translation/transcription ratio. Excluding 1% of genes in this way makes the boundary line less sensitive to measurement errors and to outlier genes, some of which are highlighted. The predicted boundary of the depleted region (red line) is according to the theory introduced later in this article. Technical constraints explain the absence of genes at low transcription and translation rates (region marked "not sampled"). *S. cerevisiae* data (**a**) from Weinberg et al.[29]. *M. musculus* (**b**) and *H. sapiens* (**c**) data from Eichhorn et al.[30]. *E. coli* data (**d**) from Li et al.[3]. **e** Transcription and translation rates of 3744 *E. coli* genes (gray dots) and of 7624 synthetic constructs (red dots) of Kosuri et al.[38]. The apparent negative correlation between transcription and translation rates in this dataset is due to limits in the linear range of flow cytometry measurements which leads to censoring of low and high abundance proteins[38]. **f** Figure legend summarizing the meaning of the different lines on **a–e**. See also Supplementary Fig. 2

**Table 1 The intercept k of the boundary of the depleted region varies over two orders of magnitude across the studied organisms**

| Organism | Measured $k$ | Predicted $k$ | $\beta_p^{max}(\times 10^3)$ | $\alpha_p$ | $\sum \beta_m (\times 10^4)$ | $c_{vO}$ |
|---|---|---|---|---|---|---|
| *S. cerevisiae* | 1.1 ± 0.1 | 1.1 ± 0.3 | 7.1 ± 0.7 | 1.3 ± 0.2 | 30 ± 15 | 0.10 ± 0.01 |
| *E. coli* | 14 ± 3 | 13 ± 4 | 11 ± 1 | 1.9 ± 0.3 | 2.0 ± 1 | 0.25 ± 0.01 |
| *M. musculus* (3T3) | 44 ± 3 | 29 ± 8 | 3.0 ± 0.3 | 0.04 ± 0.01 | 2.5 ± 1.2 | 0.3 ± 0.03 |
| *H. sapiens* (HeLa) | 66 ± 4 | 60 ± 17 | 5.1 ± 0.5 | 0.05 ± 0.01 | 1.4 ± 0.7 | 0.3 ± 0.03 |

$k$ can be predicted from the maximal translation rate $\beta_p^{max}$ [protein $mRNA^{-1}h^{-1}$], the protein decay rate $\alpha_p$ [$h^{-1}$], the total transcriptional output $\sum \beta_m$ [$mRNA \, h^{-1}$], and the noise floor $c_{vO}$ using Eq. (5). The measured $k$ is defined by having 99% of genes with $\beta_p/\beta_m > k$. Uncertainties represent standard errors determined from 3744 to 9770 genes depending on the organism

this set are essential to cellular viability, as indicated by knockout experiments (Supplementary Fig. 2b). If one removes essential genes from the data, the boundary of the depleted region shifts up and tightly fits the rest of the genes with a higher intercept, $k = 44 \pm 9$. This higher intercept for non-essential genes is predicted by the theory introduced below (Supplementary Fig. 2b, Methods).

A depleted region is also found when we estimate transcription and translation from two proteomic and mRNAseq datasets in *H. sapiens* and *M. musculus* (Supplementary Fig. 2c, d). Finally, we observe the depleted region when plotting transcription burst rate against translational burst size[36,37] inferred from single cell protein abundance measurements[17] (Supplementary Fig. 2e).

We assessed the statistical significance of the depleted region by shuffling transcription and translation rates while conserving the distributions of protein abundance and translation rates (Supplementary Fig. 2f–h). We find that none of the $10^4$ shuffled datasets show a comparable depleted region in Crick space (equal or smaller number of genes with $\beta_p/\beta_m < k$, $p < 10^{-4}$).

### Genes can mechanistically achieve high transcription and low translation.

A possible explanation for the depleted region is that a (possibly yet unknown) biochemical constraint prevents high transcription combined with low translation. To test for this possibility, we re-analyzed measurements by Kosuri et al.[38] on synthetic genes that provided a wide range of transcription and translation rates. In that study, GFP was expressed in *E. coli* under the control of 114 promoters and 111 Ribosomal Binding Sites (RBSs) of varying strengths. Relative abundance of the GFP mRNA and protein was then quantified by mRNAseq and flow cytometry.

The transcription and translation rates of the synthetic constructs largely overlap with those of *E. coli* genes (Fig. 2e, Supplementary Fig. 2i). However, in contrast to *E. coli* genes, a large fraction of the synthetic constructs achieve a combination of high transcription and low translation rates. 25% of the synthetic genes fall in the depleted region seen for endogenous *E. coli* genes. In *S. cerevisiae* as well, 32% of synthetic promoters from the library of Sharon et al.[39] fall in the depleted region (Supplementary Fig. 2j).

This observation supports the conclusion that the biochemistry of gene expression can achieve high transcription and low translation in principle. In support of this argument, there are indeed examples of such genes in the depleted region for all four organisms. These include the ribosome maturation factor *rimM* in *E. coli*, the amino-acid response regulator *GCN4* in *S. cerevisiae*, and the iron homeostasis protein *Fth1* in *M. musculus* and *H. sapiens* (Fig. 2a–d). Thus, the results in this paper concern ~99% of the genes, with the remaining ~1% requiring additional analysis (see discussion for suggested effects for these genes).

### At constant protein abundance, increasing transcription increases both precision and cost of gene expression.

Because biochemical constraints do not seem to explain the lack of genes combining high transcription with low translation, we asked whether evolutionary trade-offs might explain it.

One could hypothesize that cells avoid combining high transcription with low translation in order to minimize the cost of mRNA synthesis. In *S. cerevisiae* growing in rich medium, the fitness cost of mRNA is $c_m \sim 10^{-9}$ per transcribed nucleotide (Methods)[21]. Synthesizing a non-beneficial mRNA of length $l_m$ leads to a growth rate penalty $\Delta f_m$ that is linear with the transcription rate[21], $\Delta f_m = c_m l_m \beta_m$. For a typical mRNA of length $l_m = 1300$ nucleotides transcribed at a rate $\beta_m = 30$ mRNA h$^{-1}$, the fitness cost of transcription is thus $c_m \beta_m l_m \simeq 4 \times 10^{-5}$ per hour (Fig. 3a, Methods), which is selectable[18,21]. The cost of mRNA is also selectable in *E. coli*[20,22].

In addition to their cost, high transcription rates also have benefits in reducing the noise[13–16]. Increasing the transcription rate while keeping protein abundance fixed should therefore decrease stochastic fluctuations in protein abundance.

To test if this prediction holds genome-wide across the diversity of chromosomal context and promoters, we use measurements of cell-to-cell variations in protein abundance in *S. cerevisiae*[17] and *E. coli*[40]. Cell-to-cell variations in protein abundance can be quantified by the coefficient of variation (CV). We determine contours of the CV as a function of transcription and translation rate using Gaussian smoothing (Methods), and compare these to contours of protein abundance. Both in *S. cerevisiae* (Fig. 3b) and *E. coli* (Supplementary Fig. 3a), the CV decreases with increasing transcription and decreasing translation on each equi-protein line. The CV mainly scales with transcription, as predicted by theory[15] (Methods).

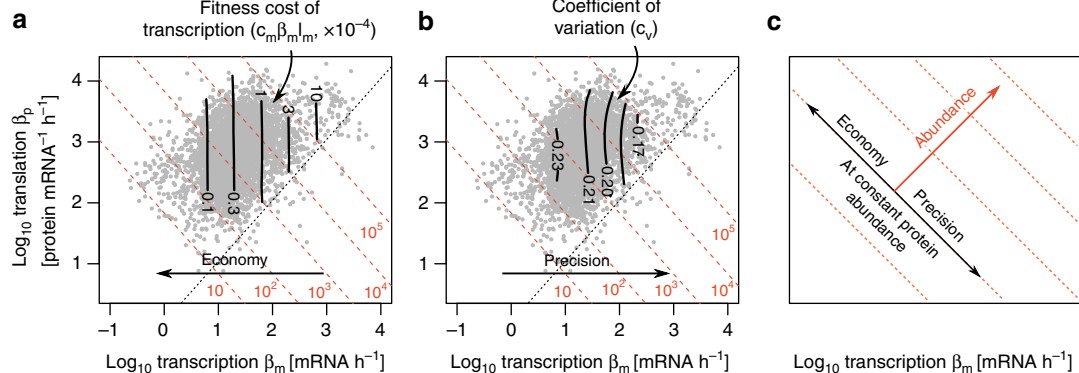

**Fig. 3** Increasing transcription at constant protein abundance increases transcription cost and decreases stochastic fluctuations in protein abundance. *S. cerevisiae* rates from Weinberg et al.[29]. Diagonal dotted lines are lines of constant protein abundance from 10 to $10^5$ proteins per cell. **a** The loss of fitness $c_m \beta_m l_m$ due to transcription (black lines) is linear in the transcription rates $\beta_m$ and the mRNA length $l_m$. The linear factor $c_m$ (introduced in the next section) rescales transcription fluxes [nt h$^{-1}$] into fitness loss [h$^{-1}$]. In *S. cerevisiae*, $l_m = 1300$ nt, $c_m = 10^{-9}$ nt$^{-1}$. **b** Coefficients of variation (CV, black lines) scale with transcription rates. We applied 2D Gaussian smoothing on CVs (Methods). *S. cerevisiae* data: CVs from Newman et al.[17], ribosome profiling and mRNAseq data from Weinberg et al.[29]. **c** Precision in gene expression increases with transcription whereas protein abundance depends both on transcription and translation. Thus, at a given protein abundance, increasing transcription increases the precision of gene expression at the expense of higher transcription costs. See also Supplementary Fig. 3

Hence, transcription and translation rates impact both gene expression precision and mRNA economy. At a given protein abundance, high translation/transcription ratios lead to economy but also higher gene expression noise, whereas low translation/transcription ratios yield high precision at the expense of higher mRNA cost (Fig. 3c). The lack of genes combining high transcription and low translation could be explained by this trade-off: for genes located in the depleted region, the benefits of increased precision may be smaller than transcriptional costs.

**The precision-economy trade-off and the noise floor explain the depleted region of Crick space.** To quantitatively test whether a trade-off between precision and economy can explain the depleted region, we developed a minimal mathematical model of the fitness cost and benefit of transcription and translation (Fig. 4). The model has two main predictions: first that the optimal ratio between translation and transcription rates $\beta_p/\beta_m$ is set by the ratio of transcription cost per mRNA molecule $C$ and the gene's sensitivity to noise $Q$ (defined below). The second prediction is an analytical formula for the boundary of the

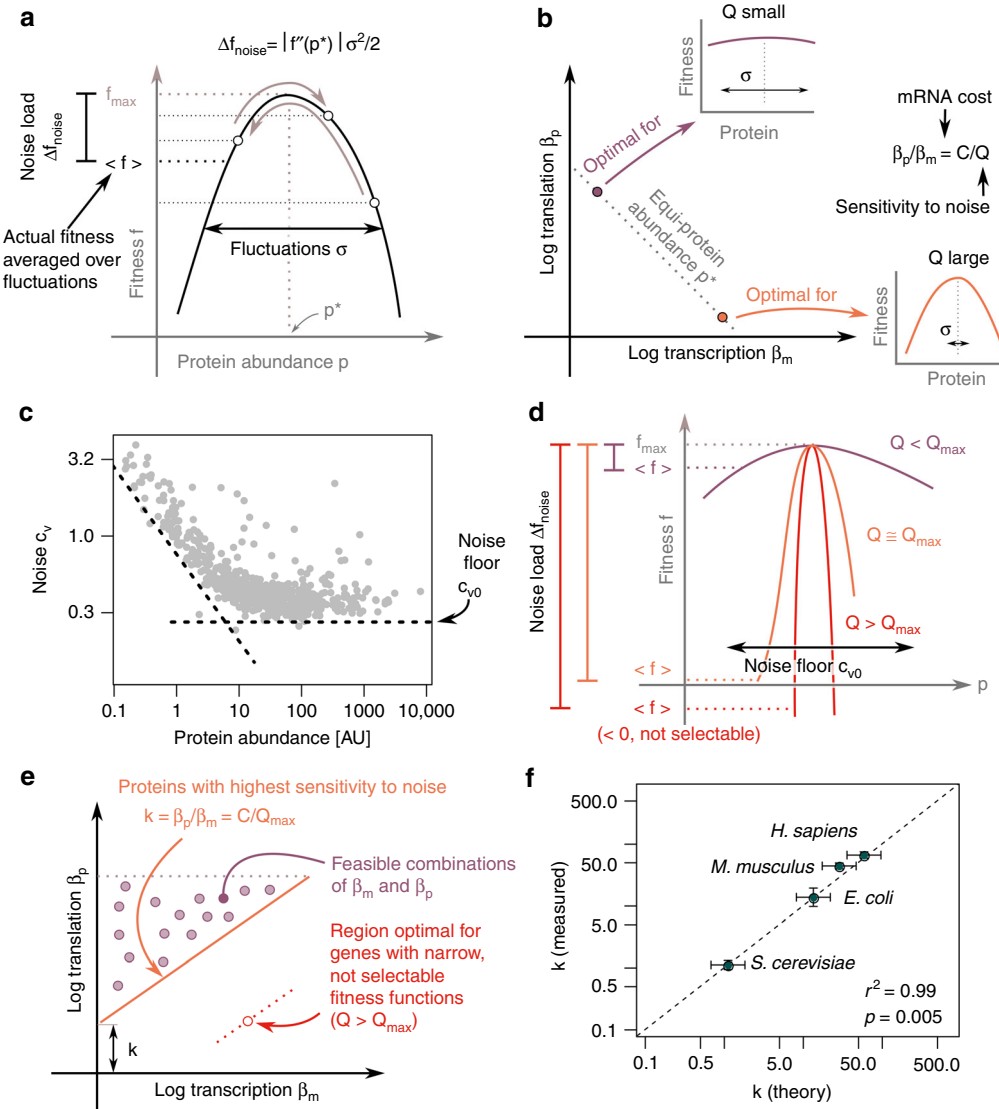

**Fig. 4** A trade-off between precision and economy can explain the depletion of genes combining high transcription with low translation. **a** The noise load $\Delta f_{noise}$ is the loss of fitness due to stochastic fluctuations in protein abundance. **b** The optimal $\beta_p/\beta_m$ depends on the transcription cost per mRNA ($C$) and on how noise sensitive ($Q \sim |f''(p^*)|$) the gene is. Noise-sensitive genes have narrower fitness functions ($|f''(p^*)|$ large). For a given $p^*$, higher transcription $\beta_m$ decreases fluctuations in protein abundance. Genes that are sensitive to noise ($Q$ large) should thus have low translation/transcription ratio. On the other hand, precision is less critical for genes with flat fitness functions ($Q$ small). These genes should have higher $\beta_p/\beta_m$ to keep transcription costs low. **c** The precision of gene expression is limited by the noise floor $c_{v0}$. Protein abundance and CV data re-plotted from Taniguchi et al.[40] The noise floor is also found in the *E. coli* measurements of Silander et al.[41] (Supplementary Fig. 4b). **d** Genes with fitness functions narrower than the noise floor $c_{v0}$ have negative average fitness ($\langle f \rangle < 0$). Genes with negative fitness are not selectable. Thus, the noise floor $c_{v0}$ prevents the selection of narrow, noise-sensitive fitness functions. **e** The maximal, selectable noise sensitivity $Q_{max}$ determines the position $k$ of the boundary of the depleted region. Proteins lying on the boundary have maximal noise sensitivity $Q = Q_{max}$. Feasible combinations of transcription and translation correspond to genes that are less sensitive to noise $Q < Q_{max}$. **f** The precision-economy trade-off theory predicts the position $k$ of the depleted region in organisms from bacteria to mammals. Error bars represent 95% confidence intervals. The *p*-value is computed using Pearson's test on 4 organisms (two-sided). The correlation between measurements and theory is robust to varying the fraction of genes used in defining the depleted region (Supplementary Fig. 4e). See also Supplementary Fig. 4

depleted region—the lower bound $k$ on the $\beta_p/\beta_m$ ratio—based on fundamental parameters.

To determine the optimal $\beta_p/\beta_m$ ratio under the precision-economy trade-off, we first model how the $\beta_p/\beta_m$ ratio affects mRNA economy and precision. We then model how mRNA economy and precision affect fitness. Finally, we determine an analytical expression for the optimal $\beta_p/\beta_m$ ratio.

To compute how the $\beta_p/\beta_m$ ratio affects economy, we model the fitness cost of transcription[21] by the linear function $\Delta f_m = c_m l_m \beta_m$ where $l_m$ is the (pre-)mRNA length and $c_m$ is the fitness penalty per transcribed nucleotide. $c_m$ can be estimated from the growth rate $\mu$ and the total transcriptional output $\sum \beta_m$ if we assume that non-beneficial mRNA are transcribed at the expense of mRNAs which code for beneficial proteins. If a total of $\sum \beta_m l_m$ nucleotides are transcribed in a cell, the average fitness contribution of each nucleotide is $\mu/\sum \beta_m l_m$. This is also the fitness lost per nucleotide of transcribing a non-beneficial mRNA at the expense of a beneficial mRNA. Thus, the fitness cost per transcribed nucleotide is

$$c_m = \frac{\mu}{\sum \beta_m l_m}. \tag{1}$$

This provides $c_m \sim 10^{-9}$ per nucleotide in *S. cerevisiae* which agrees well with experimental measurements mentioned above (Methods).

Although the cost of transcription is typically smaller than the cost of translation[18,20,21], the cost of translation needs not be modeled here. To see why, note that the cost of translation of a gene depends on how many proteins are made, regardless of whether the proteins are synthesized from few mRNAs or many mRNAs. Thus, provided that $p$ protein copies are needed, the trade-off is determined by transcription, i.e., whether many or few mRNAs are made to supply the $p$ copies. The relevant cost is hence the cost of transcription.

To see how $\beta_p/\beta_m$ and precision affect fitness, we consider a protein of abundance $p$. The protein contributes a quantity $f(p)$ to the organism's fitness (Fig. 4a). Because protein abundance fluctuates around the average expression $\langle p \rangle = p^*$, the cell does not experience the maximum fitness $f_{\max}$ but rather a lower average fitness $\langle f(p) \rangle$. The fitness lost due to stochastic fluctuations in protein abundance $\Delta f_{\text{noise}}$ is called the noise load[24,25] (Fig. 4a). By expanding the fitness function $f$ to second order and averaging over fluctuations in protein abundance, we can compute the noise load:

$$\begin{aligned}
\Delta f_{\text{noise}} &= f_{\max} - \langle f(p) \rangle \\
&= f_{\max} - \langle f_{\max} + f'(p^*)(p - p^*) \\
&\quad + \tfrac{1}{2} f''(p^*)(p - p^*)^2 \rangle \\
&= \tfrac{1}{2}|f''(p^*)|\sigma^2.
\end{aligned} \tag{2}$$

Noise load increases with the curvature of the fitness functions $|f''(p^*)|$ and with noise $\sigma$.

To find how $\beta_m$ and $\beta_p$ affect fitness through the precision of gene expression, we note that $\beta_m$ and $\beta_p$ affect the noise level $\sigma^2$ in a well-characterized way. Theory and experiments[15–17,40] indicate that the variance of protein abundance is given by

$$\sigma^2 \simeq p^2 \left( \frac{1}{p} + \frac{\alpha_p}{\beta_m} + c_{v0}^2 \right) \tag{3}$$

where $\alpha_p$ is the protein decay rate and $c_{v0}$ is the noise floor (Methods). The noise floor is the minimal amount of cell-to-cell variation in protein abundance in clonal populations[17,40–42].

We can now determine the optimal transcription and translation rates $\beta_m$ and $\beta_p$ that minimize the combination of

the transcription cost $\Delta f_m$ and of the noise load $\Delta f_{\text{noise}}$ (Methods),

$$\frac{d\left(c_m \beta_m l_m + |f''(p^*)|\sigma^2/2\right)}{d\beta_m} = 0 \Rightarrow \beta_m = \sqrt{\frac{\alpha_p \alpha_m}{C} Q p^*} \Rightarrow \frac{\beta_p}{\beta_m} = \frac{C}{Q} \tag{4}$$

where $C = c_m l_m \alpha_m = c_m l_m \beta_m/m$ quantifies the cost of transcription per mRNA molecule $m$ for this gene, and $Q = \frac{1}{2}|f''(p^*)|p^*$ is the gene's sensitivity to noise. Genes with narrower fitness functions (larger $|f''(p^*)|$) are more sensitive to noise because the noise kicks protein abundance farther from optimum. The sensitivity of a gene to noise also depends on protein abundance $p^*$ because the noise generally scales with protein abundance $\sigma^2 \sim p^*$.

The model therefore predicts a relationship between $\beta_p/\beta_m$ ratios and the shape of fitness functions (Fig. 4b). Broad fitness functions should have large $\beta_p/\beta_m$ because genes with broad fitness functions are not sensitive to noise. For those, high precision provides little benefit, and it is best to maximize economy by lowering transcription. On the other hand, genes with narrow fitness functions are sensitive to noise. For those, requirements of high precision to keep the noise load low dominate the cost of transcription. Genes with narrow fitness functions should therefore have small $\beta_p/\beta_m$ ratios.

We compared the model prediction for the curvature of the fitness function near its peak to recent measurement of fitness functions of 21 genes in *S. cerevisiae*[43]. We find that the curvatures predicted from $\beta_p/\beta_m$ are within an order of magnitude of the measured curvatures without any fitting parameters (Supplementary Fig. 4a). Predictions and measurements correlate positively ($r = 0.39$). A shuffling test suggests that the agreement between the predictions and measurements is unlikely due to chance ($p = 0.04$, Methods). However, variability in the experimental measurements precludes a conclusive comparison.

To find a lower bound $k$ on $\beta_p/\beta_m$ and explain the boundary of the depleted region, we note that there is a limit to how small the noise in gene expression can be. This limit, called the noise floor $c_{v0}$, is revealed by measurements of cell-to-cell variation of protein abundance in clonal populations[17,40–42] (Fig. 4c, Supplementary Fig. 4b, c). The cause for the noise floor is a current research topic[12] and it has been proposed that it is due to extrinsic noise[40] or larger transcriptional burst size of high abundance proteins[42]. The noise floor puts an upper bound $Q_{\max}$ on how noise-sensitive genes can be: if a gene had a fitness function narrower than this limit ($Q > Q_{\max}$), the noise load would dominate the benefit of expressing the gene (Fig. 4d), leading to negative fitness. Because genes with $Q > Q_{\max}$ cannot be selected for, all expressed genes must satisfy $Q < Q_{\max}$. The boundary of the depleted region $\beta_p/\beta_m = k$ hence corresponds to genes with highest sensitivity to noise $Q_{\max}$ (Fig. 4e). The depleted region $\beta_p/\beta_m < k$ corresponds to transcription and translation rates that are optimal for genes with fitness functions too narrow given the noise floor.

To find $k$, we substitute $Q = Q_{\max}$ in Eq. (4) for the optimal $\beta_p/\beta_m$ ratio and rewrite $k$ in terms of the noise floor and other cell biology constants (Methods):

$$k = \frac{C}{Q_{\max}} = \frac{\beta_p^{\max} c_{v0}}{\sqrt{\alpha_p \sum \beta_m}} \tag{5}$$

where $\sum \beta_m$ is the combined transcriptional output of all genes.

This expression provides an intuition for the cellular parameters which set the boundary of the depleted region of the Crick space. For example, a larger noise floor $c_{v0}$ raises the boundary because there is less benefit in having high

transcription. An increased transcriptional output $\sum \beta_m$ lowers the boundary because individual mRNAs are less costly, allowing increased precision in gene expression.

This formula for $k$ accurately predicts the boundary of the depleted region of Crick space (Fig. 4f; red lines on Fig. 2) despite the fact that $k$ varies by nearly two orders of magnitude between organisms (Table 1). Thus, the depleted region can be explained in terms of fundamental parameters such as the noise floor, maximal translation rate, total transcription output, and mean protein decay rate. Individual cellular constants alone cannot accurately predict $k$ (Supplementary Fig. 4d). Neither can $k$ be predicted by noise alone without considering economy, such as hypothesizing that the depleted region is made of all $\beta_m$ and $\beta_p$ for which increasing transcription provides little extra precision relative to the the noise floor (Methods).

## Discussion

We find that the distribution of genes in Crick space is not random: genes combining high transcription and low translation are depleted. Such combinations of high transcription and low translation can be achieved with synthetic gene constructs[38]. Therefore, mechanistic constraints cannot explain this depletion. We explain the depletion by a trade-off between precision and economy: increasing transcription at constant protein abundance diminishes stochastic fluctuations, but at a fitness penalty due to the cost of transcription. High transcription rates are therefore optimal for genes that are sensitive to noise whereas low transcription rates are well suited for genes that can tolerate high noise (Fig. 5a). A quantitative model of this trade-off predicts the curvature of the fitness function for each gene and quantitatively explains the boundary of the depleted region.

Combinations of high transcription and low translation are achievable by the gene expression machinery, as evidenced by the thousands of synthetic reporter constructs in the depleted region, and the ~1% natural genes in the depleted region. The depleted region is also easily reachable by mutations from naturally occurring genes—for example, it typically takes a single mutation to turn a strong RBS into a weak RBS[44], and single mutations in promoters can strongly affect transcription[45]. This strengthens the hypothesis that the depleted region is due to selection. We evaluate other hypotheses to explain the depleted region in the Supplementary Discussion.

Ninety nine percent of the genes lie above the predicted boundary of the depleted region. The remaining 1% fall in the depleted region. For some of these genes, evidence from focused studies indicates low translation rates and/or translation control, such as the *E. coli* outliers *rimM* and *trmD*[46], *HAC1* in *S. cerevisiae*[47], and *Fth1* in *M. musculus* and *H. sapiens*[48]. One explanation for such outlier genes is that they face constraints beyond the precision-economy trade-off. For example, the amino-acid response regulator *GCN4* in *S. cerevisiae* is strongly transcribed and poorly translated in rich medium. But under amino-acid imbalance conditions, a general decrease in protein translation triggers derepression of *GCN4* translation through a translation reinitiation mechanism involving short upstream open reading frames[49]. This regulatory mechanism couples *GCN4* synthesis to translation stress. It also bypasses transcription, which could allow for rapid upregulation. Such considerations of regulatory couplings or speed might overshadow precision-economy-based limits for certain genes, especially genes responsible for survival upon changes in the environment[50]. We consider other aspects of gene regulation beyond precision and economy in the Supplementary Discussion.

The distribution of genes in Crick space is bounded above by the maximal translation rate ($10^{3.6}$–$10^4$ proteins h$^{-1}$), and below by the boundary of the depleted region. The position of each gene in this space is determined, in the present picture, by the curvature of its fitness function. Genes with a narrow fitness function are most sensitive to noise, and are predicted to lie near the boundary of the depleted region. Genes with a broader fitness function are predicted to lie farther above this boundary. This prediction suggests an experimental test by measuring the curvature of the fitness function and comparing to the prediction. Accurate measurements of fitness functions can be performed by titrating protein concentration experimentally and measuring fitness. A recent experiment measured such fitness functions for 85 genes in *S. cerevisiae*[43]. While we found a statistical agreement between the predicted and measured curvatures of fitness functions, the measurement errors were too large to permit a meaningful comparison to the present predictions (Supplementary Fig. 4a). Further experiments to measure fitness functions can test whether transcription and translation rates predict fitness curvature near the peak. Alternatively, transcription and translation could be manipulated to quantify the effect of precision and economy on fitness. Such experiments are challenging because state-of-the art assays can detect fitness changes of ~1% which is bigger than the fitness changes visible to natural selection ($10^{-8}$ to $10^{-4}$ h$^{-1}$ depending on the species).

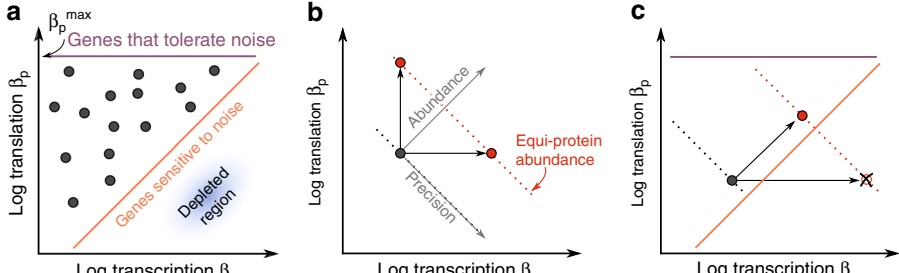

**Fig. 5** The precision-economy trade-off suggests rules for choosing transcription and translation rates and for selecting regulatory strategies. **a** Due to the precision-economy trade-off, low $\beta_p/\beta_m$ is preferred for genes that are sensitive to noise. In log-log scale, low $\beta_p/\beta_m$ corresponds to a diagonal line. On the other hand, high $\beta_p/\beta_m$ is preferred for proteins that tolerate noise. Because translation rates have an upper limit $\beta_p^{\max}$, genes with highest $\beta_p/\beta_m$ are found on a horizontal line. **b** Regulatory strategies that lead to the same protein abundance differ in how they impact precision. Upregulating transcription simultaneously increases protein abundance and precision. On the other hand, upregulating translation increases protein abundance while decreasing precision. Transcription control is thus advantageous assuming that precision is desirable. **c** When protein abundance changes by a large amount, pure transcription regulation can put the gene in the sub-optimal, depleted region. This can be avoided by co-regulating transcription and translation. See also Supplementary Fig. 5

Beyond optimizing transcription and translation under a precision-economy trade-off, an additional reason why genes may lie farther above the boundary of the depleted region is the possibility that noise is beneficial for some genes. This occurs for example in cases of bet hedging where a gene product brings little or no fitness advantage at present, but expression is maintained in case conditions change so that the gene product becomes important. In such cases, theory and experiments have shown that a wide cell–cell variation in protein level can be beneficial[51–55]. Such genes expressed for possible future needs are predicted to lie far above the boundary of the depleted region. This prediction is in agreement with the finding of stress genes at relatively high positions above the boundary, and of essential genes closer to the boundary (Supplementary Fig. 5a, b, Supplementary Table S2, Supplementary Data 1).

The theory has further testable predictions. By expressing a protein from different synthetically produced combinations of translation and transcription rates, one should find that there is an optimal translation/transcription ratio. Upon changing growth conditions such that the protein becomes even more important for growth, the optimal transcription rate should increase whereas the optimal translation rate should show little change.

The finding that essential, high-precision genes are located close to the boundary of the depleted region has implications for synthetic circuit design. If a protein needs to have a specific abundance for the circuit to function properly, that protein should be expressed using a promoter and ribosomal binding site that puts it close to the boundary of the depleted region. In other words, the optimal design should have $\beta_p/\beta_m = k$ with a value of $k$ appropriate to the organism (Table 1). On the other hand, if the circuit is insensitive to the exact concentration of that protein, the protein should be expressed using a weaker promoter and stronger RBS to save transcriptional resources. Combinations of strong transcription (promoters, enhancers) with weak translation (RBS and so on) should be avoided because they incur high transcriptional cost with no extra precision benefit.

The present approach can also help interpret the mode of regulation when the abundance of a protein needs to change. Increasing a protein level can be done by increasing transcription, translation, or both. Studies in several organisms indicate that transcription regulation is more prevalent and strong than translation regulation for most genes[3,28,32,33,56]. The present theory provides a possible explanation for this observation (Fig. 5b). Transcription regulation increases protein abundance and at the same time decreases noise. Translation regulation will increase noise. Thus transcription control is advantageous assuming that precision is more desirable than economy. Precision could be more desirable than economy for genes which become key to growth upon a change in condition, such as amino-acids biosynthesis genes upon a sudden amino-acid depletion (Supplementary Fig. 5j). For these genes, fluctuations leading to low expression would be penalizing. This penalty can be mitigated by increasing transcription. The relatively rare cases of strong translation regulation may be due to considerations of faster response time, or to cases where it is beneficial to reduce precision, such as in bet hedging[51]. One interesting case is when proteins need to be upregulated from a very-low to a very-high level. Geometric considerations rule out a purely transcriptional regulation, because this will put the gene into the depleted region; instead, a combined transcription and translation upregulation is predicted (Fig. 5c).

The present findings suggest that translation/transcription ratios are determined to some extent by rules, such as precision-economy trade-offs. Rules have been proposed in the past to explain features of the complexity of biological systems[57–59]. Patterns such as empty regions of the Crick space can help define the rules, similar to the way empty regions of morphospaces in animal morphology or single-cell gene expression can be used to infer potential evolutionary tasks and trade-offs[60–62]. It would be interesting to discover additional rules for gene expression in order to interpret the evolved design of cells and to improve engineering of synthetic circuits.

## Methods

**Central dogma rates and steady-state mRNA & protein abundance**. Using a commonly used formalism[4,26], gene expression can be modeled as a dynamic system. The system has two variables, $m$ and $p$ which represent the number of mRNAs and proteins per cell.

mRNAs are transcribed at a rate $\beta_m$ [mRNA/h] and decay at a rate $\alpha_m$. Proteins are translated at a rate $\beta_p$ [protein mRNA$^{-1}$ h$^{-1}$)] and decay at a rate $\alpha_p$.

The protein decay rate $\alpha_p$ [h$^{-1}$] combines the effect of 1. degradation $\alpha_{deg}$ [h$^{-1}$] and 2. dilution by cell growth and division which takes place at rate $\mu$ [h$^{-1}$]. Thus $\alpha_p = \alpha_{deg} + \mu$[64]. This consideration also applies to mRNA decay rates. However, mRNA decay is typically much faster than the cell cycle time (Supplementary Table 1). As a result, mRNA decay rate is set by the degradation rate.

Using these reaction rates, we can write the dynamics of $m$ and $p$ as

$$\frac{dm}{dt} = \beta_m - \alpha_m m \tag{6}$$

$$\frac{dp}{dt} = \beta_p m - \alpha_p p. \tag{7}$$

To compute steady-state mRNA and protein abundance, we set $dm/dt = dp/dt = 0$. Solving for $m$ and $p$ yields expressions for steady-state mRNA and protein abundance as function of the central dogma reaction rates

$$m = \frac{\beta_m}{\alpha_m} \tag{8}$$

$$p = \frac{\beta_m \beta_p}{\alpha_m \alpha_p}. \tag{9}$$

**Data sources and cellular constants in four organisms**. *S. cerevisiae:* We obtained the processed Reads Per Kilobase per Million (RPKMs) of the mRNAseq and RP experiments of Weinberg et al.[29] from GEO (GSE75897). We used the RiboZero mRNAseq experiment. Experimental measurements estimate $N_m = 60,000$ mRNA copies per cell[65] cellular volume at 37 μm³ (BNID100430[66]). Given a protein concentration[67] of $3 \times 10^6$ μm$^{-3}$, we estimate that there are $N_p \simeq 1.1 \times 10^8$ proteins per cell. We used a cell division time of 99 min (BNID101310). Given that the median protein half-life (excluding the dilution factor) is 45 min[68], the typical protein decay rate is $\alpha_p = 60(\log(2)/45 + \log(2)/99) = 1.34$ h$^{-1}$. Eser et al.[69] estimated the typical mRNA decay rate at $\alpha_m = 5.1$ h$^{-1}$. Multiple experiments found a noise floor $c_{v0} \simeq 0.1$[16,17,70].

*E. coli:* From the sequence reads archive, we downloaded the mRNAseq reads and RP reads from the experiments Li et al.[3] performed in rich medium (mRNAseq: SRR1067773, SRR1067774, RP: SRR1067765, SRR1067766, SRR1067767, SRR1067768). We obtained the *E. coli* genome sequence and transcriptome annotation from NCBI (accession NC_000913.3).

All genome mappings were performed using Bowtie2 in local alignment mode. We discarded all technical reads as well as reads that mapped against non-coding RNAs, defined as transcripts marked as ncRNA, rRNA, or tRNA in the genome annotation. Remaining reads were mapped to transcripts marked as CDS in the genome annotation. RP reads were mapped to coding transcripts after trimming the first and last 5 codons to remove the effect of translation initiation and termination. Reads that mapped equally well to multiple loci were assigned to one of the loci at random. We then computed RPKMs per gene. Because reproducibility between runs was high, we combined reads from all runs for subsequent analyses.

The resulting mRNA abundances and protein synthesis rates estimates were highly correlated with those computed by Li et al.[3] ($r^2 > 0.99$). Differences could be due to the updated genome version we used (NC_000913.3 vs NC_000913.2 in the original analysis of Li et al.), differences in the aligner (we used Bowtie2 while Li et al.[3] used Bowtie), and other differences in the implementation of the bioinformatics pipeline. Repeating all the *E. coli* analyses using the protein synthesis rates from Supplementary Table 1 and mRNA abundance Supplementary Table S4 of Li et al.[3] leads to minor changes in the exact position of genes in Crick space and supports all conclusions presented in this article.

There are 1380 mRNAs per cell (BNID100064). Cell volume is $\simeq 1\,\mu m^3$ (BNID100004). Assuming a protein concentration[67] of $3 \times 10^6\,\mu m^{-3}$, there are about $3 \times 10^6$ proteins per cell. Doubling time was measured by Li et al.[3] at 21.5 min, which puts the growth rate at $\mu = 60 \log(2)/21.5 = 1.93\,h^{-1}$. RP RPKMs correlate well with protein abundance[3]. This suggests that protein decay mainly takes place through dilution. Thus, protein degradation can be neglected in estimating protein decay for most proteins: $\alpha_{\text{deg}} = 0 \Rightarrow \alpha_p = \alpha_{\text{deg}} + \mu = \mu$. The median mRNA half-life is 2.8 min[71], which corresponds to decay rate $\alpha_m = 14.9\,h^{-1}$. The noise floor $c_{v0}$ was measured to be about 0.25: $0.27 \pm 0.01$ in Taniguchi et al.[40], $0.22 \pm 0.01$ in Silander et al.[41].

*M. musculus*: We downloaded the processed RPKMs of the mRNAseq and RP experiments of[30] from GEO (GSE60426).

Given a 3T3 cell volume of $V = 2000\,\mu m^3$[26] and a protein concentration of $3 \times 10^6$ per $\mu m^3$[67], there are $N_p \simeq 6.0 \times 10^9$ proteins per cell[67]. Measurements suggest that there are around $N_m = 180,000$ mRNAs per 3T3 cell[26]. Following experimental measurements[26], we used a cell cycle time of 24 h and a protein half-life of 48 h (after removing the effect of cell division[26]). These numbers correspond to a growth rate $\mu = 0.03\,h^{-1}$ and a degradation rate $\alpha_{\text{deg}} = 0.01\,h^{-1}$, which puts the protein decay rate at $\alpha_p = \mu + \alpha_{\text{deg}} = 0.04\,h^{-1}$. Friedel et al.[72] found a median mRNA decay rate of $\alpha_m = 0.14\,h^{-1}$ in 3T3 cells, which is the value we used here. Another study[26], also in 3T3 cells, measured a median decay rate of $\alpha_m = 0.08\,h^{-1}$, a value for which the predicted boundary is also in good agreement with rates measurements (Supplementary Fig. 2k). To our knowledge, the noise floor $c_{v0}$ hasn't been measured in mouse. We therefore used the noise floor from the closest organism in evolutionary terms, namely *H. sapiens*.

*H. sapiens*: We downloaded the processed RPKMs of the mRNAseq and RP experiments of[30] from GEO (GSE60426).

Given a cellular volume of $V = 2500\,\mu m^3$ (BNID103725) and a protein concentration[67] of $3 \times 10^6/\mu m^3$, there are about $N_p = 7.5 \times 10^9$ proteins per cell. A division time of 22 h (BNID109393) corresponds to a growth rate $\mu = 0.03\,h^{-1}$. The protein degradation rate measurements of Cambridge et al.[73] found $\alpha_{\text{deg}} = 0.02\,h^{-1}$. This puts the effective protein decay rate at $\alpha_p = \mu + \alpha_{\text{deg}} = 0.05\,h^{-1}$. We could not find direct measurements of the number of mRNAs per HeLa cell $N_m$. A back of the envelope calculation puts $N_m$ in the $10^5$–$10^6$ range[67]. This is consistent with recent smFISH and RNAseq measurements which found $10^5$ mRNAs in MIN6 and $10^6$ mRNAs in liver cells[74]. Starting with the 180,000 mRNAs per 3T3 cells[26] and assuming that mRNA content scales with cell volume, we estimate $N_m \simeq 180,000 \times 2500/2000 = 225,000$ mRNA per HeLa cell. The accuracy of the predicted boundary of the depleted region is robust to halving or doubling $N_m$ ($112,500 < N_m < 450,000$, see Supplementary Fig. 2l, m). Gregersen et al.[75] measured mRNA half-lives in (human) HEK293 cells and found a median half-life of 11.4 h, which we used to compute the mRNA decay rate ($\alpha_m = 0.06\,h^{-1}$). Another study measured a median half-life of 5 h in human B-cells (BL41)[72], a value for which the predicted boundary of the depleted region is also in good agreement with experimental data (Supplementary Fig. 2n). Dar et al.[42] found a noise floor $c_{v0} \simeq 0.3$.

**Estimating transcription and translation from omics datasets.** For each gene $i$, we estimated the number of mRNAs per cell $m_i$ from the total number of mRNAs per cell $N_m$ from the per-gene mRNAseq RPKM $r_i$ data as

$$m_i = N_m \frac{r_i}{\sum_j r_j}. \tag{10}$$

At steady-state, mRNA abundance $m$ is the ratio of the transcription rate $\beta_m$ to the mRNA decay rate $\alpha_m$[4]. We thus estimated the transcription rates of each gene as

$$\beta_{m,i} = m_i \alpha_m \tag{11}$$

where $\alpha_m$ is the median mRNA decay rate (Supplementary Table 1). Finally, we estimated translation rates $\beta_p$ by combining three numbers: the total number of proteins per cell $N_p$, the protein decay rate $\alpha_p$, and the gene's RP RPKM $s_i$ of gene $i$. The number of proteins synthesized per time unit is $N_p \alpha_p$. A fraction $s_i / \sum_i s_i$ of this protein synthesis flux is translated from mRNAs $m_i$ of gene $i$. We estimate the translation rate $\beta_{p,i}$ (expressed per mRNA copy per cell) by dividing the protein synthesis of each gene by the mRNA copy number $m_i$:

$$\beta_{p,i} = \frac{N_p \alpha_p}{m_i} \frac{s_i}{\sum_j s_j}. \tag{12}$$

Estimates of $N_p$ and $\alpha_p$ are provided in Supplementary Table 1.

To address the concern that gene-to-gene variations in the translation elongation rate may bias our estimates, we compared protein synthesis rates computed from protein abundance and decay data $p\alpha_p$ measured by mass-spectrometry[26] to the protein synthesis rate estimated from ribosomal density $\beta_p m$[29] on the same cell line (Mouse 3T3). We find that protein synthesis rates correlate well (Supplementary Fig. 1e, $r = 0.70$, $p < 10^{-15}$ at Pearson's test on more than 3000 genes). This agreement is probably due to the fact that although elongation rates vary from gene to gene, the initiation rates vary more (3-fold versus 1000-fold)[76]. Thus the latter tend to determine most of the variation.

To evaluate the effect of this simplification, we plot genes in 2D Crick space taking gene-specific mRNA and protein decay rates into account (Supplementary Fig. 1f). We then assign the same median mRNA and protein decay rates to all genes to re-estimate transcription and translation rates (Supplementary Fig. 1g). The gene positions in the two resulting 2D Crick spaces differ by less than 0.3 (root mean square deviation in $\log_{10}$ rates), which is $\simeq 10\%$ of the total variation (about 3 $\log_{10}$ units in transcription and translation rates). We conclude that taking into account gene-specific mRNA and protein decay rates has only a small impact on the position of genes in 2D Crick space and thus on present conclusions.

**Sequencing data processing.** We consider only genes for which the measurement error was small enough to allow accurate estimation of transcription and translation rates. Accurate estimation of these rates is difficult for low abundance mRNAs because they may only collect a handful of reads. This leads to a large uncertainty on the mRNA copy number $m$, and thus on the transcription rate $\beta_m = m\alpha_m$. How many reads per gene should we require to be confident about our estimates of transcription rates?

Estimates of mRNA abundance $m$ scale with the number of reads $n$ mapping to a given gene (relative to the gene length). We thus compute the minimum number of reads per gene needed to keep the sampling noise on $\log_{10}$ mRNA abundance below a certain threshold $\varepsilon$

$$\log_{10}\left(\frac{n+\sigma}{n}\right) < \varepsilon \tag{13}$$

where $\sigma$ is the standard deviation on $n$ due to the sampling error. We model sequencing as a Poisson process, and thus $\sigma = \sqrt{n}$. Substituting this expression for $\sigma$ into Eq. (13), we compute the minimal number of reads necessary to control for a given error $\varepsilon$ on $\log_{10}$ mRNA abundances:

$$n > \left(\frac{1}{10^\varepsilon - 1}\right)^2. \tag{14}$$

A minimum of 10 reads per mRNA is needed to keep the sampling error on $\log_{10}$ transcription rates in the $\pm 0.1$ range (Supplementary Fig. 2o). We therefore discard genes with less than 10 reads per gene, a procedure which keeps the sampling error low while keeping as many genes as possible in the analysis. Similarly, we require at least ribosomal profiling 20 reads per gene. We applied the same criteria to all four organisms. We repeated the analyses keeping only genes with at least 100 RP reads and reached the same conclusions as the one presented in the article.

In *M. musculus* and *H. sapiens*, we discarded canonical histone genes from the analysis because their mRNAs lack a polyA-tail. The polyA+ selection step of mRNAseq discriminates against these mRNAs. Consequently, the abundance of canonical histone mRNAs is underestimated by mRNAseq RPKMs, leading to aberrant (high) translation rate estimates.

**Estimating maximal translation rates.** To estimate the maximal translation rate, we ask how fast proteins can be translated $\beta_p^{\max}$ from a single mRNA in the limit where translation initiation is no longer limiting. In this regime, ribosomes follow each other closely along the mRNA. A given ribosome needs to move forward before the next one can advance. The speed at which ribosome elongate the peptide chain and how many codons each ribosome occupies on the mRNA determine how fast proteins can be synthesized.

If a ribosome occupies $L$ codons on the mRNA and $v$ codons are translated into amino-acids per second, it takes $L/v$ seconds for a ribosome to free space for the next ribosome. The maximal flux of ribosomes at given codon is thus $v/L$. This flux sets an upper bound on the translation rate: $\beta_p^{\max} = v/L$.

In *S. cerevisiae*, the elongation rate $v$ in favorable growth conditions is 10 amino-acids per second (BNID107871). Each ribosome occupies 28 nucleotides (BNID107874), so $L = 9.3$ codons. This puts the maximal translation rate $\beta_p^{\max}$ at $10^{3.6}$ proteins per hour. This bound is in good agreement with estimated translation rate in *S. cerevisiae* and other eukaryotes.

In *E. coli*, translation rate can be up to $10^4$ proteins per hour (Fig. 2). While the size of ribosomal footprints have not been determined, prokaryotes have smaller ribosomes (21 nm, BNID102320) than eukaryotes (26.5 nm, BNID111542) and so ribosomal footprints should be smaller. Assuming that ribosomal footprints are proportional to the size of the ribosome, we estimate that prokaryotic ribosomes cover 22 nt or $L = 7.3$ codons. In favorable conditions, *E. coli* can elongate up to $v = 21$ amino-acids per second (BNID100059). This leads to a maximal translation rate of $\beta_p^{\max} = 10^4$ proteins per hour.

**Testing the statistical depletion of genes in Crick space.** To test for a statistical significant depletion of genes combining high transcription with low translation, we used a randomization strategy.

First, we defined the depleted region as the sub-region of the Crick space lying below a line of slope 1 that has 1% of the genes below it. We defined the depleted region in this way because yielded a visually convincing fit between the boundary

and the data in all four organisms, using a uniform criteria. We then asked if this figure of 1% was high or low compared to chance.

To find out, we randomized transcription and translation rates. Because the main goal of gene expression is to express proteins at the right abundance, we required that randomized datasets have the same distribution of protein abundance as the original dataset. Also, we enforced the observed upper bound $\beta_{p,\max}$ on translation rates. To do so, we randomly sampled a protein abundance $p$ and a translation rate $\beta_p$ for each gene. We then computed the corresponding transcription rates $\beta_m = p\alpha_m\alpha_p/\beta_p$, with $\alpha_m$ and $\alpha_p$ the mRNA and protein decay rates reported in Supplementary Table 1. Finally, we determined what fraction of genes in the randomized dataset were found below the line of slope 1 and leaving 1% of the genes of the original dataset below it.

We repeated the procedure $10^4$ times to determine the distribution of the fraction of genes in the depleted region expected by chance. We finally estimated the $p$-value that genes avoid combining high transcription with low translation from the fraction of randomized datasets with more genes in the depleted region than the original dataset.

**Estimating transcription and translation for synthetic genes**. We compared the distribution of transcription and translation rates of *E. coli* genes to that of the synthetic constructs library of Kosuri et al.[38]. This study quantified the mRNA and protein abundance of each construct. The constructs only differed in their ribosomal binding sites and promoters. We thus assumed that they shared the same mRNA decay rates and protein decay rates. As a result, the transcription rate of each construct is proportional to mRNA abundance. Translation rates are proportional to the ratio of protein abundance to mRNA abundance.

To compare these measurements to our absolute transcription and translation rates estimates of *E. coli* genes, we assumed that the strongest promoters and RBSs of Kosuri et al.[38] yielded transcription and translation rates comparable to *E. coli*'s strongest promoters and RBS. We did so by aligning the 99th percentiles of the transcription and translation rates of the synthetic constructs to those of *E. coli* genes.

The conclusions of the comparison are robust to this assumption. For instance, even if we assume that the strongest Kosuri promoters and RBSs achieve transcription rates 10 times higher or lower than the strongest *E. coli* promoters (i.e., shifting the red cloud of Fig. 2e to the right or to the left by one unit), the high transcription–low translation region would still be covered by a sizable fraction of synthetic constructs.

**Expression for the fitness cost of transcription**. Experiments and theory suggest that the fitness cost of transcription $\Delta f_m$ scales with the transcription rates[21,22] and (pre-)mRNA length[19,77]. For a pre-mRNA of length $l_m$ and transcription rate $\beta_m$, we thus write the fitness cost of transcription $\Delta f_m$ as

$$\Delta f_m = c_m l_m \beta_m \tag{15}$$

where the constant $c_m$ rescales transcription fluxes (nucleotides per hour) into fitness units (per hour).

The exact biochemical process responsible for the cost of transcription costs is still under debate. The main candidates include the limited availability of the RNA polymerase which limits the initiation step[21] and the availability of nucleotides[18].

To estimate the proportionality constant $c_m$, we hypothesize that transcriptional resources are limiting. In this case, making one non-beneficial mRNA comes at a cost because it replaces mRNAs coding for fitness-contributing proteins, which leads to a loss of fitness. On average, the fitness contribution of a useful mRNA is $\frac{\mu}{N_m}$ where $\mu$ is the growth rate and $N_m = \sum \beta_m/\alpha_m$ is the total number of mRNAs per cell. Therefore, the fitness cost of making $m = \beta_m/\alpha_m$ mRNAs is

$$\Delta f_m = \frac{\mu}{N_m} m = \frac{\mu}{l_m N_m} l_m \frac{\beta_m}{\alpha_m}. \tag{16}$$

By identifying the terms in Eqs. (15) and (16), we see that $c_m$ can be estimated from the growth rate $\mu$, the typical pre-mRNA length $l_m$ and the total transcriptional capacity $\sum \beta_m$

$$c_m = \frac{\mu}{l_m \alpha_m N_m} = \frac{\mu}{l_m \sum \beta_m}. \tag{17}$$

$c_m$ has units of nt$^{-1}$. Alternatively, $c_m$ can also be expressed per mRNA copy (which we will use in the next section):

$$c_m = \frac{\mu}{\sum m} = \frac{\mu}{N_m} \tag{18}$$

where $N_m$ is the number of mRNAs per cell.

**Predicting mRNA cost from cellular constants in *E. coli***. In this section, we show that the expression for $c_m$ derived in the previous section predicts the fitness cost of synthesizing mRNA in *E. coli*. For this, we use the data of Kosuri et al.[38] who quantified the abundance of 10,000 different clones that express a fluorescent

protein under the control of different promoters and RBSs. As a result, different clones express the fluorescent protein at a different abundance $p$, from a different number of mRNAs $m$.

We model mRNA and protein cost $c_m$ and $c_p$ as linear penalties on the growth rate $\mu$

$$\mu = \mu_0 - c_p p - c_m m \tag{19}$$

where $\mu_0$ is the rate at which cells would grow in the absence of a fluorescent protein construct. If cells are growing exponentially for $t$ hours, the concentration $x_{m,p}(t)$ of a clone that expresses $m$ non-beneficial mRNAs and $p$ proteins is

$$x_{m,p}(t) = x_{m,p}(0)e^{\mu t}. \tag{20}$$

We can normalize $x_{m,p}(t)$ to the concentration of clones $x_{0,0}(t)$ that express $m$ and $p$ at low levels and hence don't experience a growth penalty ($\mu \simeq \mu_0$):

$$\log \frac{x_{m,p}(t)}{x_{0,0}(t)} = -tc_p p - tc_m m. \tag{21}$$

Here, we have assumed that the transformation efficiency of clones is independent of $m$ and $p$ ($x_{m,p}(0) \simeq x_{0,0}(0)$). We estimate $x_{m,p}(t)/x_{0,0}(t)$ from the ratio between the DNA counts of each clone and the DNA counts of clone that expressed low levels of GFP (prot $< 1.5 \times 10^3$ in Table S3 of Kosuri et al.[38]).

To test if mRNA cost is selectable, we perform two linear regression analyses: one regression of $\log \frac{x_{m,p}(t)}{x_{0,0}(t)}$ on $p$ alone, and one regression of $\log \frac{x_{m,p}(t)}{x_{0,0}(t)}$ on $p$ and $m$. Using the F-test for nested linear models, we find that the squared residuals for the regression on $m$ and $p$ are significantly smaller than the squared residuals for the regression on $p$ alone ($p < 10^{-15}$). Therefore, a model that accounts for mRNA and protein cost is significantly more accurate at predicting fitness than a model that account for protein cost alone. This suggests that the cost of synthesizing mRNA is selectable in *E. coli*.

The linear regression estimates of mRNA and protein cost are $tc_m = 4.0 \times 10^{-2}$ mRNA$^{-1}$, and $tc_p = 2.2 \times 10^{-6}$ protein$^{-1}$.

Since the growth time $t$ is not known precisely, we cannot determine $c_m$ and $c_p$ individually. But we can determine their ratio: $c_m/c_p \simeq 630$. The fitness cost of transcription theory introduced in the previous section (Eq. (18)) predicts

$$\frac{c_m}{c_p} = \frac{N_p}{N_m} \simeq 2100 \tag{22}$$

where $N_p$ is the total number of proteins per cell and $N_m$ is the total number of mRNAs per cell (Supplementary Table 1). Given the typical uncertainty on measurements of $N_p$ and $N_m$ (2-fold), the 95% confidence for $c_m/c_p$ ranges from 300 to 14,000. We thus find that predictions of mRNA and protein cost agree with the measurements of Kosuri et al.[38].

Finally, we test whether the theoretical estimates for $c_m$ and $c_p$,

$$c_m = \frac{\mu_0}{N_m}, c_p = \frac{\mu_0}{N_p}, \tag{23}$$

can predict the abundance of clones. While we need to know the growth time $t$ to predict clone abundance, the correlation between measured and predicted clone abundance is independent of $t$ (see Eq. (21)), which relates clone abundance to the growth time and the costs of mRNA and protein).

We find a positive correlation between predictions and measurements of clone abundance ($r = 0.67$, $p < 10^{-15}$). We set $t$ to one day ($t = 24$ h) in Supplementary Fig. 3b to illustrate the correlation.

In conclusion, the cost of mRNA synthesis in *E. coli* can be predicted from the growth rate and the total transcription output.

**Estimating the fitness cost of mRNA in *S. cerevisiae***. Here we estimate the growth penalty of transcription in *S. cerevisiae* from the measurements of Kafri et al.[21]. This study introduced a fluorescent protein construct of pre-mRNA length $l_m$ at different copy numbers $n$ in the *S. cerevisiae* genome. The protein abundance $p$ of the fluorescent protein depended on the genomic copy number of the construct, as did the transcription rate $\beta_m$.

This study found that the growth rate $\mu$ decreases linearly with the genomic copy number $n$ of the fluorescent protein construct,

$$\mu = \mu_0 - n\left(c_p p + c_m l_m \beta_m\right) \tag{24}$$

where $\mu_0$ is the growth rate of the WT strain, $c_m$ is the growth penalty of transcription (expressed per transcribed nucleotide) and $c_p$ is the protein burden (growth penalty per protein). To compare cost across growth conditions, the study normalized the growth rate $\mu$ of strains with genomic insertions of the construct to that of WT $\mu_0$ (e.g., Fig. 4b of Kafri et al.[21]):

$$\frac{\mu}{\mu_0} - 1 = -n\left(\frac{c_p p + c_m l_m \beta_m}{\mu_0}\right). \tag{25}$$

Following the notation of Kafri et al.[21], we call $s_N$ the slope of the relative growth rate $\mu/\mu_0$ as function of the genomic copy number of the construct $n$. We can write $s_N$ in terms of the fitness cost parameters $c_m$ and $c_p$:

$$s_N = \frac{c_p p + c_m l_m \beta_m}{\mu_0}. \tag{26}$$

To distinguish between the cost of protein and transcription, Kafri et al.[21] designed a second construct (DAmP) lacking a terminator, which makes the mRNA unstable. This reduces protein synthesis between 10 fold (according to protein fluorescence) and 30 fold (according to qPCR) with minimal effect on transcription[21]. Similar to $s_N$, Kafri et al.[21] defined $s_D$ as the slope of $\mu/\mu_0$ as a function of the genomic copy number $n$ of the DAmP construct. We can also write $s_D$ as a function of fitness cost parameters $c_p$ and $c_m$:

$$s_D = \frac{c_p \phi p + c_m l_m \beta_m}{\mu_0} \tag{27}$$

where the factor $\phi$ accounts for the decrease in protein synthesis of the DAmP strains.

From experimental measurements of $s_N$, $s_D$, $\phi$, $l_m$, and $\beta_m$, one can estimate the translation cost per nucleotide $c_m$:

$$\phi s_N - s_D = \frac{c_m l_m \beta_m (\phi - 1)}{\mu_0} \Rightarrow c_m = \mu_0 \frac{s_D - s_N \phi}{l_m \beta_m (1 - \phi)}. \tag{28}$$

Experiments in YPD medium[21] measured the following values: $s_N \simeq 9.0 \times 10^{-3}$, $s_D \simeq 4.1 \times 10^{-3}$, $l_m \simeq 1000$ nt/mRNA, $\phi \simeq 0.06$, $\beta_m \simeq 1300$ mRNAs h$^{-1}$, and $\mu_0 \simeq 0.42$ h$^{-1}$. Plugging in these values in Eq. (28) estimates the fitness cost $c_m$ per transcribed nucleotide,

$$c_m \simeq 1.2 \times 10^{-9} \text{ nt}^{-1}. \tag{29}$$

All parameters are known up to two significant digits, except for $\phi$ ($10 \leq \phi^{-1} \leq 30$), which is known up to one significant digit. This puts the measurement uncertainty at $\pm 1 \times 10^{-9}$ nt$^{-1}$.

**Predicting mRNA cost from cellular constants in yeast.** Plugging the cellular constants of *S. cerevisiae* from Table 1 into the formula for $c_m$ (Eq. (17)) estimates the transcription cost at $c_m = \mu_0 / \sum \beta_m l_m \simeq 1.2 \times 10^{-9}$ nt$^{-1}$. This value is in excellent agreement with the experimental measurements of Kafri et al.[21]. Such an agreement between experimental fitness parameters in the order of the $10^{-9}$ may appear surprising given the typical accuracy of biological measurements. The reason for such a good agreement is that the parameters are expressed per transcribed nucleotide whereas actual measurements were performed for full-length mRNAs of a highly transcribed *S. cerevisiae* gene. Because both the transcript length $l_m$ and the transcription rates $\beta_m$ are in the order of $10^3$ in the experiments of Kafri et al.[21], we are actually comparing fitness parameters in the order of $10^{-3}$, which are accessible experimentally.

**Plotting the coefficient of variation over Crick space.** We obtained measurements of the coefficients of variation $c_v$ of *S. cerevisiae* and *E. coli* genes from the studies of Newman et al.[17] and Taniguchi et al.[40]. To determine the contours of $c_v$ as a function of transcription and translation, we first applied a Gaussian smoother. The smoother estimates the coefficient of variation $c_v$ for given transcription $\beta_m$ and translation $\beta_p$ rates from a weighted average of genes with similar $\beta_m$ and $\beta_p$. Genes with comparable $\beta_m$ and $\beta_p$ weight more in the average than genes with very different $\beta_m$ and $\beta_p$.

Formally, we estimated $c_v(\beta_m, \beta_p)$ as a weighted average,

$$c_v\left(\beta_m, \beta_p\right) = \frac{1}{\sum_j w_j} \sum_i w_i c_{v,i} \tag{30}$$

where the Gaussian weights $w_i$ are defined as:

$$w_i = \frac{1}{\sqrt{2\pi\sigma^2}} e^{-\frac{1}{2}\left[\left(\frac{\beta_m - \beta_{m,i}}{\sigma}\right)^2 + \left(\frac{\beta_p - \beta_{p,i}}{\sigma}\right)^2\right]}. \tag{31}$$

We set the smoothing width $\sigma$ to one fifth of the data range. We only plotted contours of $c_v(\beta_m, \beta_p)$ for densely populated regions of the Crick space ($\sum w_i \geq 200$).

**Protein fluctuations as a function of the central dogma rates.** In this section, we derive an expression for the coefficient of variation $c_v$ as a function of the transcription rate $\beta_m$, the protein abundance $p$ and the protein decay rate $\alpha_p$. Following a extensive line of theoretical and experimental research[15,78], we model gene activation and inactivation as a telegraph process (Supplementary Fig. 3c). Genes are activated at a rate $k_\text{on}$ and inactivated at a rate $k_\text{off}$. Active genes synthesize mRNAs at a rate $\delta$. Messenger RNAs are translated into proteins at a rate $\beta_p$ and

degrade at a rate $\alpha_m$. At steady-state, the coefficient of variation $c_v$ on protein abundance of this stochastic process can be computed analytically[15]:

$$\begin{aligned} c_v^2 &= \left(\frac{\sigma}{p}\right)^2 \\ &= \frac{1}{p} + \frac{\alpha_p \alpha_m}{\beta_m (\alpha_p + \alpha_m)} \\ &\quad + \frac{\alpha_p \alpha_m k_\text{off} (\alpha_p + \alpha_m + k_\text{off} + k_\text{on})}{k_\text{on} (\alpha_p + \alpha_m)(\alpha_p + k_\text{off} + k_\text{on})(\alpha_m + k_\text{off} + k_\text{on})} \end{aligned} \tag{32}$$

The first term of the equation accounts for the Poisson noise on protein abundance stemming from the protein birth-death process. The second term accounts for the noise caused by translating proteins from mRNAs of low copy number. The last term models the noise caused by gene activation and inactivation and transcriptional bursting.

At present time, it is difficult to measure $k_\text{off}$ and $k_\text{on}$ genome-wide experimentally. We therefore seek a simplified, approximate expression for the coefficient of variation $c_v$ in which these parameters occur only implicitly through the transcription rate $\beta_m$. Note that $\beta_m$ and $k_\text{on}$, $k_\text{off}$ are related to each other,

$$\beta_m = \delta P_\text{on} = \delta \frac{k_\text{on}}{k_\text{on} + k_\text{off}} \tag{33}$$

where $\delta$ is the transcription rate when the gene is in the on state, and $P_\text{on}$ is the fraction of time when the gene is active.

*E. coli:* With a median half-life of 2.5 min[71], mRNAs decay much faster than proteins ($\alpha_m \gg \alpha_p$). In addition, protein decay $\alpha_p$ is mainly set by the cell division time (20 min or longer)[79], which is slow compared to gene inactivation which takes place at the time-scale of seconds[78] ($k_\text{off} \gg \alpha_p$). In this regime, we can approximate the analytical expression for the coefficient of variation (Eq. (32)) as:

$$\left(\frac{\sigma}{p}\right)^2 \simeq \frac{1}{p} + \alpha_p \left(\frac{1}{\beta_m} + \frac{k_\text{off}}{k_\text{on}(k_\text{off} + k_\text{on})}\right). \tag{34}$$

The gene activation rate $k_\text{on} \simeq 10$ h$^{-1}$ is largely constant across *E. coli* promoters, in contrast to $k_\text{off}$ which determines the transcription rate $\beta_m$[78]. Using Eq. (33) which relates the transcription rate $\beta_m$ to gene (in-)activation rates $k_\text{on}$ and $k_\text{off}$, we can rewrite the coefficient of variation in terms of $\beta_m$,

$$\left(\frac{\sigma}{p}\right)^2 \simeq \frac{1}{p} + \frac{\alpha_p}{\beta_m} + \alpha_p \frac{1 - \beta_m/\delta}{k_\text{on}} \tag{35}$$

where $\delta \simeq 800$/h and $k_\text{on} \simeq 10$/h[78].

The Poisson noise term $1/p$ is typically negligible compared to the two other terms. The small mRNA copy number noise term $\alpha_p/\beta_m$ dominates at low $\beta_m$ (Supplementary Fig. 3d). This term is consistent with the observation that protein noise initially decreases with protein abundance (Fig. 4c, Supplementary Fig. 4b, c), and that protein noise decreases with transcription (Fig. 3b).

The third term (gene activation noise) becomes dominant for large $\beta_m$ (Supplementary Fig. 3d). Because the third term is almost a constant for physiological values of $\beta_m$ (Supplementary Fig. 3d), we ask whether it could explain the noise floor found in the single cell experiments of Taniguchi et al.[40] and Silander et al.[41].

In the growth conditions of Taniguchi et al.[40], the doubling time was 150 min, which implies a protein decay rate $\alpha_p = 0.28$ h$^{-1}$. Plugging this $\alpha_p$ in Eq. (35), the third term is about 0.03, which is two-fold below the noise floor $c_{v0}^2 = 0.07$ observed in the measurements of Taniguchi et al.[40] (Fig. 4c). In Silander et al.[41] grew *E. coli* in M9 + 0.2% arabinose, a condition in which $\alpha_p \simeq 0.45$ h$^{-1}$[80]. With this $\alpha_p$, the third term is about 0.04, which is comparable to the noise floor $c_{v0}^2 \simeq 0.05$ in the measurements of Silander et al.[41] (Supplementary Fig. 4b).

We conclude that gene activation noise may explain the noise floor in the measurements of Silander et al.[41]. In the experiments of Taniguchi et al.[40], gene activation noise is too small to explain the noise floor. There, the noise floor may be explained by other factors such as as extrinsic noise[9,40]. Independently of the specific cause of the noise floor, we model it as a phenomenological constant $c_{v0}^2 \simeq 0.06$ inferred from the measurements of Taniguchi et al.[40] and Silander et al.[41] (or $c_{v0} \simeq 0.25$). This yields an expression for the noise that is accurate with both datasets,

$$\left(\frac{\sigma}{p}\right)^2 = \frac{1}{p} + \frac{\alpha_p}{\beta_m} + c_{v0}^2. \tag{36}$$

*S. cerevisiae, H. sapiens, M. musculus:* In eukaryotes, an approximate expression for the coefficient of variation can also be derived, but is slightly more complicated because the separation of time-scales is less clear than in *E. coli*: messenger RNAs decay typically faster than proteins, but not by a full order of magnitude. A more realistic, data-driven assumption is

$$\alpha_m = q \alpha_p \tag{37}$$

with $q \simeq 3$ (Supplementary Table 1). Measurements in *S. cerevisiae*[65] and *H. sapiens*[42] suggest that gene activation dynamics are much faster than protein decay,

$$k_{\text{on}} + k_{\text{off}} \gg \alpha_p. \tag{38}$$

Under these assumptions, we can approximate the analytical expression for the coefficient of variation (Eq. (32)) as

$$\left(\frac{\sigma}{p}\right)^2 \simeq \frac{1}{p} + \frac{\alpha_p}{\beta_m} \frac{q}{1+q} \left(1 + \frac{\beta_m k_{\text{off}}}{k_{\text{on}}(k_{\text{off}} + k_{\text{on}})}\right). \tag{39}$$

Using Eq. (33) which expresses transcription $\beta_m$ as function of gene (in)activation parameters $k_{\text{on}}$, $k_{\text{off}}$, $\delta$, we can eliminate $\beta_m$ from the last term:

$$\left(\frac{\sigma}{p}\right)^2 \simeq \frac{1}{p} + \frac{\alpha_p}{\beta_m} \frac{q}{1+q} \left(1 + \frac{\delta}{k_{\text{off}}} (1 - P_{\text{on}})^2\right). \tag{40}$$

We also eliminate $\delta$ by introducing the transcriptional burst size $b$, which is the average number of mRNAs that are synthesized each time the gene is activated:

$$b = \frac{\delta}{k_{\text{off}}}. \tag{41}$$

Plugging Eq. (41) for $b$ into Eq. (40) for $c_v^2$, we obtain:

$$\left(\frac{\sigma}{p}\right)^2 = \frac{1}{p} + \frac{\alpha_p}{\beta_m} \phi \tag{42}$$

where

$$\phi = \frac{q}{1+q} \left(1 + b(1 - P_{\text{on}})^2\right) \tag{43}$$

accounts for gene activation dynamics.

Except for highly expressed genes, the transcriptional burst size $b$ is typically small ($b \simeq 1$)[12]. For $q = 3$ (Supplementary Table 1), varying $P_{\text{on}}$ across its full range causes $\phi$ to vary only between 0.75 and 1.5 (Supplementary Fig. 3e). Hence, in the worst case, neglecting gene activation dynamics by setting $\phi = 1$ would result in a 1.5-fold error on $c_v^2$. We conclude that neglecting gene activation dynamics by setting $\phi = 1$ in Eq. (42) for $c_v^2$ results in a reasonable approximation of the coefficient of variation for most genes, except for highly expressed genes.

For highly expressed genes, experiments in *S. cerevisiae*[17] and *H. sapiens*[42] found a noise floor. This noise floor might occur when the transcription rate exceeds the maximal rate of gene activation[42]. In this scenario, high transcription rate can only be achieved by increasing the burst size $b$[12,42]. Since $b$ is not a constant in the large $\beta_m$ regime, we rewrite $b$ as in terms of the transcription rate $\beta_m$. To do so, we combine Eq. (33) which expresses $\beta_m$ in terms of the kinetic rates of gene activation ($k_{\text{on}}$, $k_{\text{off}}$, $\delta$) and Eq. (41) which defines the burst size $b$ in terms of $\delta$ and $k_{\text{off}}$ to find

$$b = \frac{\delta}{k_{\text{off}}} = \frac{\beta_m}{P_{\text{on}} k_{\text{off}}}. \tag{44}$$

Plugging in this expression for $b$ in Eq. (42) eliminates the burst size:

$$\left(\frac{\sigma}{p}\right)^2 = \frac{1}{p} + \frac{q}{1+q} \frac{\alpha_p}{\beta_m} + \frac{q}{1+q} \frac{(1 - P_{\text{on}})^2 \alpha_p}{k_{\text{off}} P_{\text{on}}}. \tag{45}$$

Neglecting the $1/p$ term which is in the order of $10^{-3}$ or smaller, we can get an expression for the noise floor $c_{v0}$ by taking the limit of large $\beta_m$. In this limit, the second term in Eq. (45) vanishes and $c_v^2$ approaches

$$c_{v0}^2 \simeq \frac{q}{1+q} \frac{(1 - P_{\text{on}})^2 \alpha_p}{k_{\text{off}} P_{\text{on}}}. \tag{46}$$

Plugging in gene activation parameter values typical for *H. sapiens*[42] ($P_{\text{on}} = 0.18$, $k_{\text{off}} = 2\,\text{h}^{-1}$) and setting $q = \alpha_m/\alpha_p = 3$, $\alpha_p = 0.05\,\text{h}^{-1}$ (see Table 1) puts the noise floor $c_{v0}$ at 0.27, a value comparable to experimental observations[42] ($c_{v0} \simeq 0.3$). Hence, the noise floor on protein abundance could occur when transcription rates saturate gene activation kinetics. In this case, substituting Eq. (46) for $c_{v0}^2$ into Eq. (45) leads to:

$$c_v^2 = \left(\frac{\sigma}{p}\right)^2 \simeq \frac{1}{p} + \frac{\alpha_p}{\beta_m} + c_{v0}^2. \tag{47}$$

This final expression is identical to the one we previously derived for *E. coli* (Eq. (36)). It would also hold if the noise floor was caused by a mechanism different from the transcriptional saturation of gene activation dynamics, such as extrinsic noise. Note that we neglected the $q/(1 + q)$ term of Eq. (45). This is because the resulting approximation error on $c_v^2$ would be less than 50% (Supplementary Fig. 3e). Because $\beta_m$ scales inversely with $c_v^2$ (Eq. (47)), neglecting

gene activation kinetics implies at most a 50% error on $\beta_m$, or equivalently a 0.2 error on $\log_{10}\beta_m$. This is small compared the dynamic range of transcription rates which vary over 2–3 order of magnitudes.

**Expression for the optimal translation/transcription ratio**. We consider a protein of abundance $p$. The protein contributes a quantity $f(p)$ to the organism's fitness. The cost of transcription $\Delta f_m$ is linear in the transcription rate $\beta_m$, the (pre-) mRNA length $l_m$ and the fitness cost of transcription per nucleotide $c_m$,

$$\Delta f_m = c_m l_m \beta_m. \tag{48}$$

The overall fitness is thus $F = f(p) - \Delta f_m$. The fitness function $f(p)$ reaches its maximum $f_{\text{max}}$ at $p = p^*$ (Fig. 4a). Expanding $f(p)$ around the optimum $p = p^*$ to second order, the overall fitness becomes

$$\begin{aligned} F(p, \beta_m) &\simeq f_{\text{max}} + f'(p^*)(p - p^*) \\ &\quad + \tfrac{1}{2} f''(p^*)(p - p^*)^2 - \Delta f_m. \end{aligned} \tag{49}$$

$f'(p^*) = 0$ since $f(p)$ reaches its optimum $f_{\text{max}}$ at $p = p^*$. $f''(p^*)$ is the curvature of the fitness function at its maximum. It is a negative number which characterizes how narrow the fitness function is.

Because protein abundance fluctuates around $p^*$, the cell doesn't experience the maximum fitness $f_{\text{max}}$ but rather a lower average fitness $\langle f \rangle$. Averaging $F$ over fluctuations in protein abundance, we obtain

$$\begin{aligned} \langle F(p, \beta_m) \rangle &= f_{\text{max}} + \tfrac{1}{2} f''(p^*)\sigma^2 - c_m l_m \beta_m \\ &= f_{\text{max}} - \Delta f_{\text{noise}} - \Delta f_m \end{aligned} \tag{50}$$

where $\sigma^2$ is the variance of protein abundance fluctuations (Fig. 4a). The curvature times this variance, $-\tfrac{1}{2} f''(p^*)\sigma^2$, is the noise load $\Delta f_{\text{noise}}$, the fitness lost due to the stochastic fluctuations in protein abundance[24,25] (Fig. 4a). It is a positive quantity because the curvature $f''(p)$ is negative at the maximum $p = p^*$.

We seek the transcription rate $\beta_m$ that maximizes fitness. For this purpose we note that $\beta_m$ affects the noise level $\sigma^2$ in a well-characterized way. Theory and experiments of gene expression noise[15–17,40] indicate that variance of protein noise is given by

$$\sigma^2 \simeq p^2 \left(\frac{1}{p} + \frac{\alpha_p}{\beta_m} + c_{v0}^2\right) \tag{51}$$

where $\alpha_p$ is the protein decay rate and $c_{v0}$ is the noise floor due to extrinsic noise[40] or the larger transcriptional burst size of high abundance proteins[42] (see previous section). We can now solve for the $\beta_m$ that maximizes fitness, by finding $\frac{d\langle F \rangle}{d\beta_m} = 0$. The optimal translation/transcription ratio rate $\beta_p/\beta_m$ satisfies

$$\frac{\beta_p}{\beta_m} = \frac{2 c_m l_m \alpha_m}{-f''(p^*) p^*} = \frac{C}{Q} \tag{52}$$

where we have used $p^* = \frac{\beta_m \beta_p}{\alpha_m \alpha_p}$. Note that we introduced two new variables $C$ and $Q$.

$$C := c_m l_m \alpha_m = c_m l_m \beta_m / m = \Delta f_m / m \tag{53}$$

represents the fitness cost of transcription per mRNA molecule for a gene with pre-mRNA length $l_m$.

$$Q := -\frac{1}{2} f''(p^*) p^* \tag{54}$$

is the gene's sensitivity to noise. Genes with narrower fitness functions ($f''(p^*) \ll 0$) are more noise sensitive. A gene's sensitivity to noise also depends on $p^*$ because stochastic fluctuations $\sigma^2$ scale with $p^*$ (Eq. (51)).

From the expression for the optimal $\beta_p/\beta_m$ (Eq. (52)), we see that genes dominated by comparable requirements of precision and economy are predicted to share the same translation/transcription ratio. Genes with narrow fitness function require more precision and thus have lower translation/transcription ratios (Fig. 4b). Higher transcription cost per mRNA molecule—due to longer mRNAs $l_m$ or rapid mRNA turn-over $\alpha_m$ or a scarcity of nucleotides leading to increased cost $c_m$—shifts the balance towards higher ratios.

Note that the translation/transcription ratio does not depend on translation cost, although this cost is typically larger than transcriptional cost[18,20,21]. This is because we assumed that, for a given protein abundance, translation cost are the same if the proteins are synthesized from few or many mRNAs.

**Predicting the boundary of the depleted region of Crick space**. In this section, we ask what is the predicted offset of the line that forms the boundary of the depleted region, namely the constant $k$ such that $\beta_p/\beta_m > k$ for all genes. We provide estimates based on known fundamental parameters of cell biology suggested by the theory.

To estimate $k$, we note that the precision-economy theory predicts that low $\beta_p/\beta_m$ occur for genes with narrow fitness functions (Eq. (52), Fig. 4b). But the noise floor $c_{v0}$ (Fig. 4c) sets a limit on how narrow fitness functions can be: for fitness functions that are too narrow given the noise floor, average fitness is negative (Fig. 4d). Such fitness functions cannot be selected for in evolution. We can therefore estimate the maximal transcription rate by determining the largest, selectable fitness function curvature, and then compute the optimal transcription rate for that function. The curvature should be small enough that the smallest, unavoidable protein fluctuations (set by the organism's noise floor $c_{v0}$) do not dominate the fitness benefit of expressing the protein. In other words, the fitness benefit of expressing the protein $f_{max}$ should be larger than the noise load $\Delta f_{noise}$:

$$f_{max} - \Delta f_{noise} > 0 \Rightarrow f_{max} - \frac{1}{2}|f''(p^*)|\sigma^2 > 0. \tag{55}$$

Here we neglected mRNA cost because for proteins with narrow fitness functions, it is small compared to the noise load $\Delta f_{noise}$.

If fitness is mainly set by the growth rate $\mu$, the fitness contribution $f_{max}$ of a gene cannot be larger than the growth rate: $f_{max} < \mu$. In addition, fluctuations cannot be smaller than the noise floor, $\sigma/p^* > c_{v0}$[17,40–42]. From these two considerations, we can compute an upper bound on the noise sensitivity $Q$,

$$Q = \frac{1}{2}|f''(p^*)|p^* < \frac{f_{max}}{p^* c_v^2} < \frac{\mu}{p^* c_{v0}^2}. \tag{56}$$

By substituting this upper bound in Eq. (52) for the optimal $\beta_p/\beta_m$, we find an upper bound on $\beta_m^{max}$ on transcription rates

$$\beta_m < \beta_m^{max} = \frac{1}{c_{v0}}\sqrt{\frac{\alpha_p \alpha_m \mu}{C}}. \tag{57}$$

We now consider the (hypothetical) protein expressed at maximal transcription $\beta_m^{max}$ and maximal translation $\beta_p^{max}$. This protein has highest protein abundance $p^*$. It also has narrowest fitness function (narrower fitness are not selectable due to the noise floor), and thus highest noise sensitivity $Q_{max}$. We can plug Eq. (57) for $\beta_m^{max}$ into Eq. (52) for the optimal $\beta_p/\beta_m$ to find $Q_{max}$:

$$\frac{\beta_p^{max}}{\beta_m^{max}} = \frac{C}{Q_{max}} \Rightarrow Q_{max} = \frac{\sqrt{\alpha_p \alpha_m \mu C}}{\beta_p^{max} c_{v0}}. \tag{58}$$

In this expression, we can see that a higher noise floor implies that genes need to be less sensitive to noise. We now use $Q_{max}$ in the Eq. (52) for the optimal $\beta_p/\beta_m$ to find $k$,

$$\frac{\beta_p}{\beta_m} = \frac{C}{Q} > \frac{C}{Q_{max}} = k \Rightarrow k = \beta_p^{max} c_{v0}\sqrt{\frac{C}{\alpha_p \alpha_m \mu}} = \beta_p^{max} c_{v0}\sqrt{\frac{c_m l_m}{\alpha_p \mu}} \tag{59}$$

where we have used $C = c_m l_m \alpha_m$ (Eq. (53)). The fitness cost per transcribed nucleotide $c_m$ can be estimated from the average contribution of each nucleotide of each mRNA to the organism's fitness, $c_m = \mu/\sum \beta_m l_m$ (Eq. (17)). Neglecting differences in mRNA length between genes, we finally find:

$$k \simeq \frac{\beta_p^{max} c_{v0}}{\sqrt{\alpha_p \sum \beta_m}}. \tag{60}$$

Thus, the lowest translation/transcription ratio can be predicted from measurable, fundamental parameters of each organism.

In this derivation, we have assumed that the gene's contribution to fitness $f_{max}$ is smaller than the growth rate $\mu$, $f_{max} < \mu$ (Eq. (56)). For essential genes, $f_{max} \simeq \mu$. For non-essential genes, we can use a tighter upper bound on $f_{max}$:

$$f_{max} < \rho \mu \tag{61}$$

with $0 < \rho < 1$. For example, if deleting a gene decreases fitness by 1% or less, we have $\rho = 0.01$. By repeating the derivation, we find

$$k \simeq \frac{\beta_p^{max} c_{v0}}{\sqrt{\alpha_p \rho \sum \beta_m}}. \tag{62}$$

Thus, for non-essential genes ($\rho \ll 1$), the predicted boundary of the depleted region has higher intercept.

In the Supplementary Methods, we consider alternative theories to explain the boundary of the depleted region, such as the existence of a power-law scaling of mRNA noise with mRNA abundance. None of the alternative theories can explain the depleted region.

**Code availability**. We deposited the code used to produce the figures of the present manuscript at Mendeley data, https://doi.org/10.17632/2vbrg3w4p3.1 [https://data.

mendeley.com/datasets/2vbrg3w4p3/draft?a=955cbbdf-9f26-4fbb-970b-e6b4081c1f3e].

## Data availability

The data (RPKMs from RNAseq and ribosomal profiling) from which we estimated transcription and translation rates was deposited at Mendeley data, https://doi.org/10.17632/2vbrg3w4p3.1 [https://data.mendeley.com/datasets/2vbrg3w4p3/draft?a=955cbbdf-9f26-4fbb-970b-e6b4081c1f3e]. Estimated transcription and translation rates are also found at that address. In addition, downloadable tables contain extra fields (such as coefficient of variation on protein abundance fluctuations) needed to reproduce Figs. 3b and 4c.

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

## Acknowledgements

We thank Benjamin Towbin, Shalev Itzkovitz, Ron Milo, Naama Barkai, Moshe Kafri, Mihaela Zavolan, Luca Ciandrini, Nir Friedman, Yoav Voichek, Tzachi Pilpel, Idan Frumkin, Dvir Schirman, Noam Stern-Ginossar, Michal Shreberk, Dan Davidi, Daniel Goodman, Tabitha Bucher, and members of the Alon lab for scientific discussions and feedback on the manuscript.

This work was supported by the European Research Council under the European Union's Seventh Framework Program/ERC Grant agreement 249919, and the Israel Science Foundation. U.A. is the incumbent of the Abisch-Frenkel Professorial Chair. J.H. acknowledges the support of EMBO (ALTF 1160-2012), the Swiss National Science Foundation (P300P3_158472), and the Swiss Society of Friends of the Weizmann Institute.

## Author contributions

Conceptualization, J.H. and U.A.; methodology, J.H., A.M., L.K. and U.A.; formal analysis, J.H.; Writing, J.H. and U.A.; and funding acquisition, J.H. and U.A.

## Additional information

**Competing interests:** The authors declare no competing interests.

