## [Peer Review File · Nature Communications]

Reviewers' comments:

Reviewer #1 (Remarks to the Author):

In this manuscript, Hausser et al. address an important and interesting question in gene expression: given that the same protein concentration can be achieved by different combinations of transcription and translation rates, what combinations do cells actually use? They report that cells avoid the combination of a high transcription rate with a low translation rate. They propose that this phenomenon results from a tradeoff between precision and economy, because having a high transcription rate reduces gene expression noise but wastes energy for transcription. I am impressed with the phenomenon reported, but am unconvinced by the underlying mechanism they proposed to explain the phenomenon. This may be in part due to their writing style, which often lacks precise definitions or clear explanations. There is a large amount of modeling work in the paper that I believe is valuable, but the lack of clarity in writing prevents me from evaluating it critically. This is frustrating because I spent many hours trying to understand it and appreciate it, but remain confused in the end. My detailed comments follow.

1. It is well known that transcription is much less costly than translation per cell. Andreas Wagner estimated that the total transcriptional cost is about 20 times lower than the total translational cost in a cell (PMID: 15758206). So, I am not sure that the energy savings from reduced transcription are biologically significant. The present manuscript did not convince me that the transcriptional cost should be of concern to a cell. For example, I have trouble understanding the meaning of C_m in Eq. (1). C_m is the growth rate divided by the total number of nucleotides in RNA, and authors' interpretation of C_m is the per-nucleotide contribution to growth. But growth is possible not simply because of transcription. It is unclear why they attribute growth entirely to transcription. Furthermore, they say that C_m is the per nucleotide transcriptional cost, but did not say from what processes this cost arises. Is this the energy cost of RNA synthesis, energy cost of RNA synthesis plus protein synthesis, or something else? They also similarly computed C_p , per amino acid protein synthesis cost. In Eq. (18), they say $C_m/C_p = N_p/N_m = \text{number of proteins per cell} / \text{number of RNAs per cell}$. How can the costs of transcription and translation be determined not by the biochemical processes of transcription and translation, but by the numbers of products? Because this definition of C_m and C_p is key to their model, I am at loss.

2. Authors used ribo-seq data to estimate protein synthesis rates (i.e., translational initiation rates). By doing so, they made an implicit assumption that translational elongation rates are the same across all mRNAs. There is a lot of empirical evidence against this assumption (e.g., PMID: 25051069; 26656907). I wonder how much their results change if they consider this factor. In fact, translational initiation rates have been estimated using the data of protein concentration and degradation rate (PMID: 25051069). I wonder if similar results will be found if authors use those estimates of protein synthesis rates, which do not rely on making the false assumption of equal translational elongation rates.

3. Is economy the only reason for low transcription and low noise the only benefit for high transcription? Authors did not consider any alternative reason. Is it possible, for example, that low transcription is preferred because it reduces the total number of transcriptional/translational errors in proteins?

4. They argue that the depletion of the combination of a high transcription rate and a low translation rate is not due to some unknown biochemical constraint and demonstrate this by plotting the joint distribution of transcription and translation rates from artificial gene constructs. However, this is just showing that the combination of high transcription and low translation is biochemically achievable. To show that this combination is selected against, they should ideally show that this combination is NOT mutationally rare, because it is possible for some combinations to be mutationally rare such that they look depleted yet can be made in artificial constructs. I realize that it is difficult to prove that it is not mutationally rare, but it will be important to discuss this caveat.

5. The predicted bounds and measured bounds match quite well (Table 1). However, the boundary is defined by a line of slope=1 below which 1% of genes fall. Why do they choose the 1% criterion? I wonder whether their results are robust to this arbitrary criterion.

6. Page 11. Authors discuss the possibility of beneficial gene expression noise. In yeast, such genes were previously detected (PMID: 19690568). It would be important to check if they lie well above the boundary of the depleted region.
7. Page 12. In the paragraph starting "The present approach can also", author try to explain why transcriptional regulation is more prevalent than translational regulation. Authors discuss the situation when an increase in protein amount is desired and argue that transcriptional regulation can reduce noise, but it is unclear why the increased transcriptional cost is not considered. Also, when a reduction of protein amount is desired, transcriptional regulation will increase noise but reduce transcriptional cost. So, overall, under the authors' model, there should be no advantage of using transcriptional regulation over translational regulation.
8. Page 34, when discussing HeLa and 3T3 cell lines, authors argue that these cells have likely adapted to fast growth in rich medium. This is true, but it should then be made clear that whatever results they get from these cells, they do not represent cells in vivo.
9. Many definitions are missing or unclear and many assertions are not explained, making the manuscript very hard to read. Some examples are here.
 - (1) Page 2, when beta-m and beta-p are first mentioned, no definition was given. It is important to define all variables, including specifying the units.
 - (2) In the same paragraph, no explanation was given for the formula $(\beta\text{-m} * \beta\text{-p})/(\alpha\text{-m} * \alpha\text{-p})=p$. This formula is actually not trivial and relies on some assumptions. In this formula, alpha-m and alpha-p were simply said to be decay rates without clear definitions.
 - (3) Page 3, when citing ref. 31, authors say that expressing a non-beneficial gene in yeast decreases growth in proportion to transcriptional rate. Is the cost due to transcription alone or the sum of transcription and translation?
 - (4) Eq. (1) needs better justification. Why is growth attributed to transcription only?
 - (5) Page 8, Authors cite ref. 31, 44, and 82 in stating that "the cost of translation typically dominates transcriptional cost". This sentence literally means that most of the transcriptional cost is translation. In other words, the word "transcription" means both transcription and translation. This is VERY confusing. I wonder if authors actually mean this or it is a typo.
 - (6) Page 9. "The noise floor puts an upper bound Q_{\max} on how noise-sensitive genes can be: for genes with fitness functions narrower than this limit $Q > Q_{\max}$, the noise load dominates the benefit of expressing the gene (Fig. 4D), leading to negative fitness." I don't understand this sentence, especially "negative fitness", which is impossible.
 - (7) Page 10. "The predicted depleted region seems to account for about 99% of the genes. The remaining 1% fall in the depleted region." The first "depleted" should be "undepleted"?
 - (8) When "decay" is used in Methods and main text, does it include (i) both dilution caused by cell division and degradation, or (ii) degradation alone? Authors did not define clearly, and used the meaning of (i) in some cases but (ii) in other cases. For instance, the decay rate of 1.34 per hour on page 35 apparently refers to meaning (i), but the decay rate of 14.9 per hour on page 36 appears to refer to meaning (ii). They sometimes also used the term of median half-life, but seemed to actually mean decay rate. For instance, the value of 0.08 per hour was used for "median half-life" of mRNA in mouse, but half-life should not have the unit of per hour; this is the unit for decay rates. However, it is still unclear whether they were using meaning (i) or (ii) of decay rates. It is so confusing!
 - (9) Related to the above point, when the term "half-life" is used, do authors consider meaning (i) or (ii) of the above? No definition was given.
 - (10) The above is a partial list, and there are many more.

Reviewer #2 (Remarks to the Author):

In this study, Hausser et al characterized a 2-dimensional Crick space by considering both transcription and translation rates. They found that genes with fast transcription and slow translation are depleted. They explained this distribution of genes in Crick space by the trade-off

between the precision of protein dosage in a cell and the energy cost of transcription.

Alon lab has published a series of studies related to the concepts of trade-off and optimization. For example, they showed the trade-off between energy of protein synthesis and enzyme activity in a Nature paper in 2005. They also introduced the concept of Pareto optimality, which largely explains the phenotypical spaces in multiple species. This work is an important addition to these studies. I believe it is suitable to be published in Nature Communications after addressing the following comments.

1. This study focuses on rapidly growing cells. Do the conclusions of this study hold when considering other conditions? The dilution rate is high in cells under the condition of rapid growth. In *E. coli* and yeast, dilution is likely much faster than degradation. Does the reduction from 4-dimension to 2-dimension Crick space make sense in cells in the stationary stage?
2. Related to 1, the equilibrium of gene expression in log phase is only one of the agents of natural selection. The speed of switch among environments is another. A previous review paper discussed this (PMID: 17379352).
3. Whereas the trade-off between precision and economy could explain the depletion of genes with high transcription and low translation rates, it does not necessarily exclude other possibilities. For example, when a gene needs a higher protein level, mutations enhancing either transcription or translation are selected for. If mutational target size (the number of mutations that can potentially lead to elevated protein level) is greater for enhancing transcription, will we also observe the patterns in Figure 2?
4. The authors show that mechanistic coupling is unlikely to explain the distribution of genes in the Crick space in *E. coli* (Fig. 2E). Is this also true in eukaryotes where RNA modifications (e.g. m6A) and RNA binding proteins are more abundant?
5. Some genes are expressed constitutively whereas others are expressed in bursts (burst-like transcription). It has been reported that in eukaryotes histone modifications can independently regulate transcription level and noise (PMID: 22683268, 28665997). In other words, two genes with the same mRNA level may exhibit a significant difference in expression noise. How does the transcription mode affect the conclusions of this study?
6. If I understand it correctly, Q is estimated from β_p/β_m , and is compared to 21 genes previously measured in yeast. Is it possible to independently estimate dosage sensitivity from Newman et al (ref 50) or from tug-of-war experiments (PMID: 23275495), and use this dosage sensitivity to predict the position of a gene on the 2-dimensional Crick space (β_p/β_m)?
7. It is apparently not reasonable to request manipulative experiments to measure the fitness effect of expression noise or additional transcripts. The reason is that the difference in fitness that can be detected in experiments (10^{-2}) is much greater than that visible to natural selection (10^{-8} to 10^{-4} depends on species). This reason, however, may not be clear to broader readers of Nature communications. Some explanations are encouraged.

Reviewer #3 (Remarks to the Author):

In this study, Hausser et al. reanalyzed several past experimental datasets to estimate transcription and translation rates of most genes in cultured cells from several species (bacteria, yeast, mouse and human). They observe that a portion of the 2D-space defined by these rates (which they call the 'Crick space') is very poorly populated: very few genes display fast transcription and slow translation. They call this portion the 'depleted region' of the 'Crick space'. Using previous data from a library of synthetic constructs expressed in *E. coli*, they show that this region can be populated, which suggested that natural genes avoid high transcription/low translation for evolutionary reasons. The authors propose a trade-off model, where the reduced noise in expression is counterbalanced by the cost of producing RNA. They provide mathematical

expressions of this model in terms of fitness / noise relationships and fitness cost of transcription and they show that this model can predict the boundary of the depleted region.

I enjoyed reading this interesting study. While this trade-off model is simple to propose, the study is remarkable on two aspects: revealing the 'depleted region' and providing a quantitative prediction of its boundary (k) from fitness considerations. Writing is clear and precise, methods are well described (except point 5 below) and thorough. I have the following recommendations.

1. Estimation of transcription rates is made directly from decay rates (formula 7, page 37) and the paper would be stronger if:

- The gene-specific decay rate were used instead of the median value. The authors show in Fig S1E-F that using gene-specific rates has little effect on the Crick-space distribution for mouse. Since they have gene-specific decay rates for other species as well (Fig S1A-D), it would be more rigorous to use these values instead of justifying an unnecessary approximation.
- The deduced transcription rates were compared to the rates reported from studies that were specifically designed for this. What is the correlation between the rates computed and values from transient labelling experiments (refs 18, 41, 11, 66)?

2. The authors do not discuss at all an important recent literature coupling the two axes of the 'Crick space': co-translational mRNA decay (PMIDs 25768907, 27058298, 26046441, 29507394....). The authors estimate transcription rates directly from mRNA decay rates. So if fast translation is associated to mRNA stability, then we'd expect few genes in the bottom left and top right corners of the space. This does not contradict the authors conclusions but the fact that the two axes are not mechanistically independent should be discussed: how is codon optimality distributed in the Crick space? What consequence would have a change in the proportion of co-translational decay versus translation-independent mRNA decay (between genes, between conditions...)?

3. Another important point is not discussed: the consequence of transcriptional errors. At low transcription/high translation, each RNA 'error' is largely represented among protein molecules; whereas at high transcription/low translation, each error is carried only on a minority of protein molecules. This means that in one case (top left corner of the space) the sources of errors are exposed to function/selection and in the other case (depleted region) they are not. The implication of this should be discussed, especially as error-rates differ between genes. For example, how do the error rates of Gout et al. PMID 29062891 distribute in the Crick space?

4. The population of red points in Fig2E should be presented as a density plot (as for other panels). Not clear where the highest density is and what fraction maps to the depleted region, this is important.

5. A major value of the study is to accurately predict k values from the trade-off model. For this reason, the test shown in Fig S4D should be better explained: why would anyone expect "a significant correlation between these constants and measured k " as written in legend? What was done here exactly?

6. Terms m and $\alpha.m$ appear in main text page 8 without a definition. In methods, $\alpha.m$ is defined as the median decay rate, but in main text I assume that in the expression of C it is gene-specific. This must be clarified.

7. Regarding the synthetic yeast promoters data of refs 33 & 34: these promoters confer varying levels of transcription rates and noise, with presumably similar translation rate (same coding sequence). As for the synthetic *E. coli* data, what fraction of them populate the depleted region?

Reviewer #1 (Remarks to the Author):

In this manuscript, Hausser et al. address an important and interesting question in gene expression: given that the same protein concentration can be achieved by different combinations of transcription and translation rates, what combinations do cells actually use? They report that cells avoid the combination of a high transcription rate with a low translation rate. They propose that this phenomenon results from a tradeoff between precision and economy, because having a high transcription rate reduces gene expression noise but wastes energy for transcription. I am impressed with the phenomenon reported, but am unconvinced by the underlying mechanism they proposed to explain the phenomenon. This may be in part due to their writing style, which often lacks precise definitions or clear explanations. There is a large amount of modeling work in the paper that I believe is valuable, but the lack of clarity in writing prevents me from evaluating it critically. This is frustrating because I spent many hours trying to understand it and appreciate it, but remain confused in the end. My detailed comments follow.

We thank the reviewer for the feedback. We have now addressed concerns of lack of precise definitions and clear explanations (see details below).

1. It is well known that transcription is much less costly than translation per cell. Andreas Wagner estimated that the total transcriptional cost is about 20 times lower than the total translational cost in a cell (PMID: 15758206). So, I am not sure that the energy savings from reduced transcription are biologically significant. The present manuscript did not convince me that the transcriptional cost should be of concern to a cell. For example, I have trouble understanding the meaning of C_m in Eq. (1). C_m is the growth rate divided by the total number of nucleotides in RNA, and authors' interpretation of C_m is the per-nucleotide contribution to growth. But growth is possible not simply because of transcription. It is unclear why they attribute growth entirely to transcription. Furthermore, they say that C_m is the per nucleotide transcriptional cost, but did not say from what processes this cost arises. Is this the energy cost of RNA synthesis, energy cost of RNA synthesis plus protein synthesis, or something else? They also similarly computed C_p , per amino acid

protein synthesis cost. In Eq. (18), they say $C_m/C_p=N_p/N_m$ =number of proteins per cell / number of RNAs per cell. How can the costs of transcription and translation be determined not by the biochemical processes of transcription and translation, but by the numbers of products? Because this definition of C_m and C_p is key to their model, I am at loss.

We thank the reviewer for these questions, which provides us an opportunity to clarify our argument. The reviewer raises four points: (1) why should transcription costs matter if translation costs dominate? (2) how can we estimate the growth cost of mRNA c_m from the growth rate and the total number of RNA nucleotides when other processes are necessary for cell growth? (3) what biochemical process is responsible for c_m ? (4) How can RNA costs c_m and protein costs c_p be determined by the number of products as opposed to biochemical processes?

- (1) We agree that translation costs typically dominate transcription costs. However, we believe that transcription costs are more relevant than translation costs for the present purpose: to determine the optimal number of mRNAs needed to synthesize a given number of proteins. To see why, we note that, for a given gene, translation costs scale with the number of synthesized proteins, no matter how many mRNAs these proteins are synthesized from: translating 1000 proteins from 1 mRNA requires as many ribosomes and amino-acids as translating 1000 proteins from 100 mRNAs. As a result, translation costs are insensitive to the number of mRNAs used. Instead, the optimal number of mRNAs for that gene depends on transcription costs.

Transcription costs have been reported to be selectable in different model organisms (Frumkin et al., Mol Cell 2017; Kafri et al., Cell Reports 2016; Lynch & Marinov PNAS 2015).

We now clarify this subtle but important point on page 8: "Although the cost of transcription is typically smaller than the cost of translation [35,48,89], the cost of translation needs not be modeled here. To see why, note that the cost of translation of a gene depends on how many proteins are made, regardless of whether the proteins are synthesized from few mRNAs or many mRNAs. Thus, provided that p protein copies are needed, the tradeoff is determined by transcription, *i.e.* whether many or few mRNAs are made to supply the p copies. The relevant cost is hence the cost of transcription."

- (2) We agree that the cellular proliferation rate depends on more than having the right pool of mRNAs. For example, it also depends on composition of the proteome, on the sufficient availability of ribosomes, and more. What allows us to estimate the cost of transcription from the growth rate and the total number of RNA nucleotides is that the growth rate of a cell depends on the simultaneous availability of all these components. Among these cellular components, the transcriptome is a key part because it determines the protein pool. If we hypothesize that transcriptional resources are limiting, making one non-beneficial mRNA leads to losing one fitness-contributing mRNA. Displacing the fitness-contributing mRNA leads to the loss of the encoded proteins, which then leads to loss of fitness. Dividing the total fitness (estimated by the growth rate) by the total number of mRNAs estimates the average fitness contribution of single mRNA, and thus the cost of making a non-beneficial RNA. This estimate of mRNA cost accurately predicts the fitness loss due to expressing non-beneficial mRNAs measured experimentally in *S. cerevisiae* and in *E. coli* (pg 50-53).

We now state why the cost of mRNA can be estimated without explicitly taking other

cellular components needed for growth into account in the main text (pg 7) and in the methods (pg 49).

- (3) We agree that it would be satisfying to point out the exact biochemical process responsible for transcriptional costs. This point is still under debate. The main candidates include the limited availability of the RNA polymerase which limits the initiation step (Kafri et al. Cell Rep 2016) and the availability of nucleotides (Wagner 2005). However, identifying process responsible for RNA cost is not needed for our theory to hold. The theory we introduce here only requires to quantify how this cost changes with the transcription rate. As mentioned in point (2) above, the estimate we propose to RNA cost matches experimental measurements.

In the methods, we now discuss the possible biochemical process responsible for transcriptional costs and uncertainties in the current state of the art (pg 49).

- (4) We agree with the referee that the biochemical processes of transcription and translation set the costs of transcription c_m and translation c_p . These costs can be estimated from known cellular quantities such as the growth rate or the total number mRNAs and proteins based on theoretical arguments and reasonable assumptions. For an illustration, see point (2) above for example. To test the theory, cost estimates can be compared to experimental measurements in model organisms where such measurements have been collected (*E. coli*, *S. cerevisiae*). We find that cost measurements in these organisms are in good agreement with the theory (see Methods, pg 50-53).

2. Authors used ribo-seq data to estimate protein synthesis rates (i.e., translational initiation rates). By doing so, they made an implicit assumption that translational elongation rates are the same across all mRNAs. There is a lot of empirical evidence against this assumption (e.g., PMID: 25051069; 26656907). I wonder how much their results change if they consider this factor. In fact, translational initiation rates have been estimated using the data of protein concentration and degradation rate (PMID: 25051069). I wonder if similar results will be found if authors use those estimates of protein synthesis rates, which do not rely on making the false assumption of equal translational elongation rates.

We thank the reviewer for this chance to increase the rigor of our protein synthesis rates estimates. As the reviewer points out, there are variations in the elongation rate. Previous work indicate the variation is about 3-fold (Shah & Plotkin, Cell 2013).

We now repeated the analysis with the data of protein concentration and degradation rate suggested by the reviewer (Fig S1 and S2).

We compared protein synthesis rates $p \alpha_p$ computed from protein abundance p and protein decay data α_p measured by mass-spectrometry to the protein synthesis rates estimated from ribosomal density on the same cell line (Mouse 3T3). We find that protein synthesis rates correlate well ($r=0.70$, $p<10^{-15}$) considering that measurements were obtained using different methodologies and in different labs. We now add this to the SI (Fig S1E).

The agreement is probably due to the fact that although elongation rates vary from gene to gene, the initiation rates vary more (3-fold versus 1000-fold, Shah & Plotkin, Cell 2013). Thus the latter tend to determine most of the variation.

An additional study showed that ribosomal densities recover the abundance of proteins of known abundance as well as the stoichiometry of known protein complexes in *E. coli* (Li et al., Cell 2014).

Using the mass spectrometry data provides a depleted region as in the sequence data (Fig. S1F and Fig. S1G for mouse, Fig. S2D for human). There are different details for the boundary, which may be due to the different experimental biases of mass spectrometry and sequencing. To provide an additional test, we added data from a third technique, flow-cytometry, to estimate transcription and translation in yeast. We also find a depleted region, as in the case of sequencing data (Fig S2E). We conclude that the existence of a depleted region is robust to experimental technique and is also found when variation in elongation rates are included. We believe that this strengthens the main result of the paper.

We now address this point in the methods (pg 45).

3. Is economy the only reason for low transcription and low noise the only benefit for high transcription? Authors did not consider any alternative reason. Is it possible, for example, that low transcription is preferred because it reduces the total number of transcriptional / translational errors in proteins?

We thank the reviewer for this suggestion, and we now include it on page 39 of the manuscript:

“We proposed here that a tradeoff between economy and precision in protein abundance explains the depletion of genes with high transcription and low translation. However, the rate of transcription and translation could also impact protein sequence accuracy. For example, the risk that a transcriptional error propagates to many proteins could be reduced by increasing or decreasing transcription, leading to potential tradeoffs between sequence accuracy, precision and economy. At present, tradeoffs involving protein sequencing accuracy are not supported by the data. To see why, we note that changing transcription only changes the probability of error in protein sequence if the error rate depends on the transcription rate. If q is the probability that a newly transcribed mRNA contains an error, the probability that protein synthesis will proceed from an mRNA that carries an error is

$$q \beta m / (q \beta m + (1 - q) \beta m) = q$$

which does not depend on the transcription rate βm . Thus, changing transcription only changes protein sequence accuracy if the error rate depends on the rate of transcription. Measurements of error rates for thousands of genes in *S. cerevisiae* show no correlation between transcription error rates and transcription rates [28]. Variations in error rates over Crick space are found in a narrow range between 4.0×10^{-6} nucleotide⁻¹ and 6.1×10^{-6} nucleotide⁻¹, with no clear correlation with transcription, translation or the translation / transcription ratio (Fig. S3F). Thus, considerations of protein sequence accuracy are unlikely to deplete genes with high transcription and low translation rate.”

4. They argue that the depletion of the combination of a high transcription rate and a low translation rate is not due to some unknown biochemical constraint and demonstrate this by plotting the joint distribution of transcription and translation rates from artificial gene constructs. However, this is just showing that the combination of high transcription and low translation is biochemically achievable. To show that this combination is selected against, they should ideally show that this combination is NOT mutationally rare, because it is possible for some combinations to be mutationally rare such that they look depleted yet can be made in artificial constructs. I realize that it is difficult to prove that it is not mutationally rare, but it will important to discuss this caveat.

We thank the reviewer for suggesting this caveat. We now address this alternative explanation to the depleted region in the discussion (pg 11):

“Combinations of high transcription and low translation are achievable by gene expression machinery, as evidenced by the thousands of synthetic reporter constructs in the depleted region, and the ~1% of the natural genes in the depleted region. The depleted region is also easily reachable by mutations from naturally occurring genes- for example, it typically takes a single mutation to turn a strong RBS into a weak RBS [Sallis & Voigt, *Nat Biotech* 2009], and single mutations in promoters can strongly affect transcription [Yona & Gore, *Nat Comm* 2018]. This strengthens the hypothesis that the depleted region is due to selection”

5. The predicted bounds and measured bounds match quite well (Table 1). However, the boundary is defined by a line of slope=1 below which 1% of genes fall. Why do they choose the 1% criterion? I wonder whether their results are robust to this arbitrary criterion.

We now clarify why we chose the 1% criterion: Page 47 in the methods now reads “We chose to define the boundary of the depleted region so that it leaves 1% of genes below it because doing so yielded a visually convincing fit between the boundary and the data in all four organisms, using a uniform criteria.”

We also tested the robustness of our conclusions to this 1% criterion. The agreement between theoretical predictions and measurements is robust to a varying the fraction of genes in the depleted region by 10-fold (from 0.2% to 2%). We have added a new figure (Fig S4E) to illustrate this point, and refer to in the legend of Fig. 4.

6. Page 11. Authors discuss the possibility of beneficial gene expression noise. In yeast, such genes were previously detected (PMID: 19690568). It would be important to check if they lie well above the boundary of the depleted region.

We thank the reviewer for bringing the findings of Zhang et al. to our attention. We now refer to the article in our manuscript.

Using the protein abundance and noise data of Newman et al. (*Nature* 2006), Zhang et al. found that plasma-membrane-associated transporters (PMAT) have excess noise: they are noisier than expected given their protein abundance and their importance to fitness. We examined the position of these genes in Crick space and find no significant evidence that these genes lie well above the boundary of the depleted region.

This result is expected because Zhang et al. adjusted genes by both abundance and fitness (growth effect of deletion) whereas we do not adjust for fitness. When not adjusting for fitness, PMAT genes are not noisier than other genes (Newman et al., *Nature* 2006). Thus they fall about as far from the depleted region as other genes.

Following the reviewer's suggestion, we now locate other potential bet-hedging genes in Crick space. We find that these genes lie well above the boundary of the depleted region. We now address this point on page 80 of the manuscript:

“Genes potentially involved in bet-hedging strategies have high β_p / β_m . For example, in *E. coli*, oxidative stress response induces tolerance to antibiotics and persistence [Molina & Camilli, *mBio* 2018; Vega & Collins, *Nature Chemical Biology* 2012]. Persistence is a bet-hedging strategy used by bacteria to cope with unpredictable environments [Balaban &

Leibler, *Science* 2004]. Consistent with their role in regulating persistence, oxidative stress response genes have high β_p / β_m .

In *S. cerevisiae*, a single-cell screen previously identified respiration genes as potential bet-hedging genes [Levy & Siegal, *PLoS biology* 2012]. Respiration provides tolerance to glucose starvation at the expense of a slower growth rate [Gray & Werner-Washburne, *Microbio Mol Biol Rev* 2004; Albers & Gustafsson, *Appl. Environ. Microbiol.* 2007; Levy & Siegal, *PLoS biology* 2012; Basan & Hwa, *Nature* 2015]. Consistent with their possible role in hedging glucose starvation, respiration genes have high β_p / β_m (Fig. S5E)."

7. Page 12. In the paragraph starting "The present approach can also", author try to explain why transcriptional regulation is more prevalent than translational regulation. Authors discuss the situation when an increase in protein amount is desired and argue that transcriptional regulation can reduce noise, but it is unclear why the increased transcriptional cost is not considered. Also, when a reduction of protein amount is desired, transcriptional regulation will increase noise but reduce transcriptional cost. So, overall, under the authors' model, there should be no advantage of using transcriptional regulation over translational regulation.

We thank the reviewer for this comment, which allows us to clarify our argument.

In page 13, we now write: "The present theory provides a possible explanation for this observation (Fig. 5B). Transcription regulation increases protein abundance and at the same time decreases noise. Translation regulation will increase noise. Thus transcription control is advantageous assuming that precision is more desirable than economy. Precision could be more desirable than economy for genes which become key to growth upon a change in condition, such as amino acids biosynthesis genes upon a sudden amino acid depletion (Fig. S5J). For these genes, fluctuations leading to low expression would be penalizing. This penalty can be mitigated by increasing transcription. The relatively rare cases of strong translation regulation may also be due to considerations of faster response time, or to cases where it is beneficial to reduce precision, such as in bet hedging [38]."

In the case of gene down-regulation, the argument runs similar. If the change of growth condition required more proteins at increased precision, reverting back to the original condition would require less precision, suggesting that transcription can be reduced, leading to benefits of economy.

8. Page 34, when discussing HeLa and 3T3 cell lines, authors argue that these cells have likely adapted to fast growth in rich medium. This is true, but it should then be made clear that whatever results they get from these cells, they do not represent cells in vivo.

We agree and we thank the reviewer for pointing out this omission. Page 39 now reads "Hence, both HeLa and 3T3 cells are likely well adapted to the conditions in which transcription and translation were measured, namely rapid growth in rich culture medium. Note that these growth conditions may not represent the healthy environment of mammalian cells in vivo, but rather the growth conditions found in a tumor or in cell culture."

9. Many definitions are missing or unclear and many assertions are not explained, making the manuscript very hard to read. Some examples are here.

(1) Page 2, when beta-m and beta-p are first mentioned, no definition was given. It is important to define all variables, including specifying the units.

We thank the reviewer for suggesting these improvements which will make our manuscript readable. We now systematically introduce units the first time a variable is introduced (pg 2), as well as in Fig. 1 and Fig. 2.

*(2) In the same paragraph, no explanation was given for the formula $(\beta_m * \beta_p)/(\alpha_m * \alpha_p)=\rho$. This formula is actually not trivial and relies on some assumptions. In this formula, α_m and α_p were simply said to be decay rates without clear definitions.*

On pg 2, we now refer to a new methods section 'Central dogma rates and steady-state mRNA and protein abundance'. In this section (found on pg 41), we now give explicit definitions of all the rates and derive the formula $(\beta_m * \beta_p)/(\alpha_m * \alpha_p)=\rho$.

(3) Page 3, when citing ref. 31, authors say that expressing a non-beneficial gene in yeast decreases growth in proportion to transcriptional rate. Is the cost due to transcription alone or the sum of transcription and translation?

In Ref 31, the authors determined cost due to both transcription and translation. They could tease apart these two contributions by using different conditions (eg limitations of amino acids versus nucleotides) and reporters with different RNA stability. They conclude that transcript costs are sizable and vary linearly with transcription rate.

We now rephrased the sentence on pg 3:

“Experiments in *S. cerevisiae* [35] indicate that expressing a non-beneficial mRNA penalizes the growth rate in proportion to the transcription rate (Methods)”

(4) Eq. (1) needs better justification. Why is growth attributed to transcription only?

On page 7, we now write: “ μ can be estimated from the growth rate μ and the total transcriptional output $\sum \beta_m$ if we assume that non-beneficial mRNAs are transcribed at the expense of mRNAs which code for beneficial proteins.”

In the methods, on page 49, we now write: “In this case, making one non-beneficial mRNA comes at a cost because it replaces mRNAs coding for fitness-contributing proteins, which leads to a loss of fitness. On average, the fitness contribution of a useful mRNA is μ / Nm where μ is the growth rate and $Nm = \sum \beta_m / \alpha_m$ is the total number of mRNAs per cell.”

We refer the reviewer to our answer to comment 1.(4) for the rationale behind these changes.

(5) Page 8, Authors cite ref. 31, 44, and 82 in stating that "the cost of translation typically dominates transcriptional cost". This sentence literally means that most of the transcriptional cost is translation. In other words, the word "transcription" means both transcription and translation. This is VERY confusing. I wonder if authors actually mean this or it is a typo.

We thank the reviewer for pointing out the confusion. We believe the confusion stemmed from omitting a logical step in the sentence.

On page 8, we now write: “Although the cost of transcription is typically smaller than the cost of translation [35,48,89], the cost of translation needs not be modeled here. To see why, note that the cost of translation of a gene depends on how many proteins are made, regardless of

whether the proteins are synthesized from few mRNAs or many mRNAs. Thus, provided that p protein copies are needed, the tradeoff is determined by transcription, *i.e.* whether many or few mRNAs are made to supply the p copies. The relevant cost is hence the cost of transcription.”

(6) Page 9. *“The noise floor puts an upper bound Q_{max} on how noise-sensitive genes can be: for genes with fitness functions narrower than this limit $Q > Q_{max}$, the noise load dominates the benefit of expressing the gene (Fig. 4D), leading to negative fitness.” I don't understand this sentence, especially “negative fitness”, which is impossible.*

We agree with the reviewer: negative fitness cannot be selected for, which leads to an upper bound on Q_{max} . We rewrote to clarify this point. Page 9 now reads:

“The noise floor puts an upper bound Q_{max} on how noise-sensitive genes can be: if a gene had a fitness functions narrower than this limit ($Q > Q_{max}$), the noise load would dominate the benefit of expressing the gene (Fig. 4D), leading to negative fitness. Because genes with $Q > Q_{max}$ cannot be selected for, all endogenous genes must satisfy $Q < Q_{max}$.”

(7) Page 10. *“The predicted depleted region seems to account for about 99% of the genes. The remaining 1% fall in the depleted region.” The first “depleted” should be “undepleted”?*

We thanks the reviewer for this suggestion. The sentence now reads: “99% of the genes lie above the predicted boundary of the depleted region. The remaining 1% fall in the depleted region.”

(8) *When “decay” is used in Methods and main text, does it include (i) both dilution caused by cell division and degradation, or (ii) degradation alone? Authors did not define clearly, and used the meaning of (i) in some cases but (ii) in other cases. For instance, the decay rate of 1.34 per hour on page 35 apparently refers to meaning (i), but the decay rate of 14.9 per hour on page 36 appears to refer to meaning (ii). They sometimes also used the term of median half-life, but seemed to actually mean decay rate. For instance, the value of 0.08 per hour was used for “median half-life” of mRNA in mouse, but half-life should not have the unit of per hour; this is the unit for decay rates. However, it is still unclear whether they were using meaning (i) or (ii) of decay rates. It is so confusing!*

We thank the reviewer for this chance to clarify our definition of protein decay rates.

In the introduction, we now write: “Steady-state protein abundance is set by two reactions of synthesis — transcription and translation — balanced by two processes of decay – dilution and degradation of mRNAs and proteins”

In the introduction: “ α_m [1/h] and α_p [1/h] are the mRNA and protein decay rates (Methods).”

We added a new section in the methods, 'Central dogma rates and steady-state mRNA and protein abundance' which defines mRNA and protein decay explicitly:

“The protein decay rate α_p [1/h] combines the effect of 1. degradation α_{deg} [1/h] and 2. dilution by cell growth and division which takes place at rate μ [1/h]. Thus $\alpha_p = \alpha_{deg} + \mu$ [15]. This consideration applies to mRNA decay rates. However, mRNA decay is typically much faster than the cell cycle time (Table 1).”

In the legend of Table S1 which lists the rates, we now write: “protein decay rate $\alpha_p = \alpha_{deg} + \mu$ [h^{-1}]”

On page 43, in the paragraph on *E. coli*, we now explicitly refer to our definition of protein decay, which applies consistently to all analyses of our article: “Ribosome profiling RPKMs correlate well with protein abundance [45]. This suggests that protein decay mainly takes place through dilution. Thus protein degradation can be neglected in estimating protein decay for most proteins: $\alpha_{deg} = 0 \Rightarrow \alpha_p = \alpha_{deg} + \mu = \mu$ ”.

We now corrected two instances where we wrote half-life instead of decay rate (highlighted on pg 43-44).

(9) Related to the above point, when the term "half-life" is used, do authors consider meaning (i) or (ii) of the above? No definition was given.

We have now specified whether protein half-lives refer to the combined effect of degradation and dilution, or degradation alone (see the highlights on page 41-43).

On page 41, we now write: “This consideration applies to mRNA decay rates. However, mRNA decay is typically much faster than the cell cycle time (Table 1). As a result, mRNA decay rate is set by the degradation rate.”

(10) The above is a partial list, and there are many more.

We thank the reviewer for this feedback which helped us improve the clarity of the manuscript.

Reviewer #2 (Remarks to the Author):

In this study, Hausser et al characterized a 2-dimentional Crick space by considering both transcription and translation rates. They found that genes with fast transcription and slow translation are depleted. They explained this distribution of genes in Crick space by the trade-off between the precision of protein dosage in a cell and the energy cost of transcription.

Alon lab has published a series of studies related to the concepts of trade-off and optimization. For example, they showed the trade-off between energy of protein synthesis and enzyme activity in a Nature paper in 2005. They also introduced the concept of Pareto optimality, which largely explains the phenotypical spaces in multiple species. This work is an important addition to these studies. I believe it is suitable to be published in Nature Communications after addressing the following comments.

1. This study focuses on rapidly growing cells. Do the conclusions of this study hold when considering other conditions? The dilution rate is high in cells under the condition of rapid growth. In E. coli and yeast, dilution is likely much faster than degradation. Does the reduction from 4-dimention to 2-dimention Crick space make sense in cells in the stationary stage?

We focus on growing cells because there is ample data and the tradeoffs seem clear. We now add to the paper analysis of a dataset with slower growth (depletion of aminoacids in yeast,

Fig. S2P). We find a depleted region. The boundary has a different slope, indicating perhaps a different tradeoff. Identifying this tradeoff could be an excellent follow-up study.

We now add the new Fig. S2P. We now refer to conditions of slow growth on page 39 of the manuscript.

2. Related to 1, the equilibrium of gene expression in log phase is only one of the agents of natural selection. The speed of switch among environments is another. A previous review paper discussed this (PMID: 17379352).

We thank the reviewer for this suggestion. We now refer to the article suggest by the reviewer in the discussion on page 11: “[...] This regulatory mechanism couples GCN4 synthesis to translation stress. It also bypasses transcription, which could allow for rapid up-regulation. Such considerations of regulatory couplings or speed might overshadow precision-economy-based limits for certain genes, especially genes responsible for survival upon changes in the environment (Perez-Ortin et al., *Trends in Genetics* 2007).”

3. Whereas the trade-off between precision and economy could explain the depletion of genes with high transcription and low translation rates, it does not necessarily exclude other possibilities. For example, when a gene needs a higher protein level, mutations enhancing either transcription or translation are selected for. If mutational target size (the number of mutations that can potentially lead to elevated protein level) is greater for enhancing transcription, will we also observe the patterns in Figure 2?

We thank the reviewer for this creative suggestion. We have now explored mutational target size by simulations and added this to the discussion of alternative explanations (pg 40).

In the new Methods section 'Simulating different target size of mutations affecting transcription and translation' (pg 69-70), we simulated evolution towards a given protein abundance (drawn from the natural protein abundance distribution) by mutations that affect transcription and translation rates with different mutational targets. We find that different mutation target size ratios do not create a depleted region (new Fig. S5K). The distribution in simulated Crick space is along a diagonal that is nearly orthogonal to the distribution of natural gene. We conclude that target size alone is not a viable explanation for the depleted region.

4. The authors show that mechanistic coupling is unlikely to explain the distribution of genes in the Crick space in E. coli (Fig. 2E). Is this also true in eukaryotes where RNA modifications (e.g. m6A) and RNA binding proteins are more abundant?

We now address this point in the discussion on page 38:

The quantitative model rests on the assumption that transcription and translation rates can be tuned independently. Although coupling is seen in bacteria [60] and eukarya [29], that coupling has itself evolved rather than being an absolute constraint. Studies on synthetic promoters and ribosomal binding sites in *E. coli* indicate that transcription and translation rates can be changed independently over a wide range [41].

In *S. cerevisiae* too, flow-cytometry analysis of synthetic gene constructs suggest that the eukaryotic gene expression machinery can achieve transcription and translation rates that position genes in the depleted region of Crick space (Fig. S2J).

5. Some genes are expressed constitutively whereas others are expressed in bursts (burst-like transcription). It has been reported that in eukaryotes histone modifications can independently regulate transcription level and noise (PMID: 22683268, 28665997). In other words, two genes with the same mRNA level may exhibit a significant difference in expression noise. How does the transcription mode affect the conclusions of this study?

We now discuss the effect of transcription mode and bursting in the manuscript. Page 40-41 now reads:

“Our theory focuses how on considerations of precision and economy set transcription and translation rates. Yet, precision in gene expression depends not only on the transcription rate but also on the epigenetic environment which affects the transcription mode — burst size, burst frequency [92,96]. As a result, two genes with the same transcription rate may exhibit different noise.

The transcription mode does not affect our predictions of the boundary of the depleted region of Crick space because the formula for the boundary of the depleted region (Equ. 5) does not depend on the transcriptional mode of the gene, only on the noise floor c_{v0} .

The transcription mode has a only small effect on the optimal translation / transcription ratio. The reason is that most of the noise in gene expression can be described by combining transcription, protein abundance and extrinsic noise (Equ. 36). While the epigenetic environment of individual genes can cause deviations from the formula for c_v^2 of Equ. 36, we estimate that these deviations are smaller than 50% (see section 'Expression for the variance in protein abundance as a function of the Crick rates' in the Methods). This estimate is consistent with the observation that knocking out epigenetic regulators impacts burst size and frequency by less than 30% on average [92]. A 50% error on the c_v^2 translates to a 50% error on the optimal translation/transcription ratio (see Equ. 54 and Eqn. 56). This error is small compared to the 3 orders of magnitude variation in transcription and translation rates.”

6. If I understand it correctly, Q is estimated from β_p/β_m , and is compared to 21 genes previously measured in yeast. Is it possible to independently estimate dosage sensitivity from Newman et al (ref 50) or from tug-of-war experiments (PMID: 23275495), and use this dosage sensitivity to predict the position of a gene on the 2-dimensional Crick space (β_p/β_m)?

We tried to estimate Q from knock out experiments, haploid deletions and over-expression experiments who measured the impact of dosage on fitness (Sopko, *Mol Cell* 2006 Deutschbauer, *Genetics* 2005). Estimating dosage sensitivities at sufficient accuracy proved difficult for several reasons:

1. in both datasets, the precision of fitness measurements was lower compared to Keren et al. *Cell* 2016
2. in Sopko et al., protein abundance following over-expression was unknown, which prevented estimation of dosage sensitivity
3. in the deletion experiments of Deutschbauer et al., it was unclear whether fitness far away from optimal protein abundance allows estimation of the curvature of the fitness function at optimal abundance.

These limitations are addressed by the data of Keren et al. (*Cell* 2016) which measured abundance-to-fitness curves using ~100 different promoters for each gene, in growth competitions experiments with multiple time points. We use this data to perform the analysis suggested by the reviewer. We determined dosage sensitivity (Q) for 21 genes in yeast. We then predicted dosage sensitivity from bp/bm. While we found a statistical agreement between the predicted and measured dosage sensitivities (Fig. S4A), the measurement errors were too large to permit a meaningful comparison to the present predictions.

We conclude that the theory has the potential to predict dosage sensitivity from bp/bm. Further experiments to accurately measure abundance-to-fitness curves, translation and transcription in homogeneous growth conditions are needed to test whether transcription and translation rates predict dosage sensitivity.

7. It is apparently not reasonable to request manipulative experiments to measure the fitness effect of expression noise or additional transcripts. The reason is that the difference in fitness that can be detected in experiments (10^{-2}) is much greater than that visible to natural selection (10^{-8} to 10^{-4} depends on species). This reason, however, may not be clear to broader readers of Nature communications. Some explanations are encouraged.

We thank the reviewer for this suggestion. On page 12, we now write “Alternatively, transcription and translation could be manipulated to quantify the effect of precision and economy on fitness. Such experiments are challenging because state-of-the art assays can detect fitness changes of ~1% which is bigger than the fitness changes visible to natural selection (10^{-8} to 10^{-4} depending on the species).”.

Reviewer #3 (Remarks to the Author):

*In this study, Hausser et al. reanalyzed several past experimental datasets to estimate transcription and translation rates of most genes in cultured cells from several species (bacteria, yeast, mouse and human). They observe that a portion of the 2D-space defined by these rates (which they call the 'Crick space') is very poorly populated: very few genes display fast transcription and slow translation. They call this portion the 'depleted region' of the 'Crick space'. Using previous data from a library of synthetic constructs expressed in *E. coli*, they show that this region can be populated, which suggested that natural genes avoid high transcription/low translation for evolutionary reasons. The authors propose a trade-off model, where the reduced noise in expression is counterbalanced by the cost of producing RNA. They provide mathematical expressions of this model in terms of fitness / noise relationships and fitness cost of transcription and they show that this model can predict the boundary of the depleted region.*

I enjoyed reading this interesting study. While this trade-off model is simple to propose, the study is remarkable on two aspects: revealing the 'depleted region' and providing a quantitative prediction of its boundary (k) from fitness considerations. Writing is clear and precise, methods are well described (except point 5 below) and thorough. I the following recommendations.

We thank the reviewer for this endorsement.

1. Estimation of transcription rates is made directly from decay rates (formula 7, page 37) and the paper would be stronger if:

- The gene-specific decay rate were used instead of the median value. The authors show in Fig S1E-F that using gene-specific rates has little effect on the Crick-space distribution for mouse. Since they have gene-specific decay rates for other species as well (Fig S1A-D), it would be more rigorous to use these values instead of justifying an unnecessary approximation.

- The deduced transcription rates were compared to the rates reported from studies that were specifically designed for this. What is the correlation between the rates computed and values from transient labelling experiments (refs 18, 41, 11, 66)?

We have now performed the analysis suggested by the reviewer using gene specific degradation rates. This means that there are 2-fold fewer genes due to missing data. We find essentially the same depleted region (compare Fig. S1F to Fig. S1G). In the revised paper, we now discuss this point. We show the analysis using gene-specific degradation in the SI, and use the estimate using the mean decay rate in the main figures, because it provides the same essential results, allows us to include 2-fold more genes, and avoids the concern that degradation rates are measured in a separate study than the other rates in *H. sapiens*, *E. coli* and *S. cerevisiae*.

Page 4 now reads:

“Taking into account gene-specific mRNA and protein decay rates has only a small impact on the position of genes in 2D Crick space (Fig. 1F–G, Methods).”

“Reducing to two dimensions yields a more complete picture of Crick space (mRNA and protein decay rates have typically been measured for 20% – 50% of genes) and avoids the concern that the decay rates have been measured in separate studies than the synthesis rates for *S. cerevisiae*, *E. coli* and *H. sapiens* (Methods).”

2. The authors do not discuss at all an important recent literature coupling the two axes of the 'Crick space': co-translational mRNA decay (PMIDs 25768907, 27058298, 26046441, 29507394....). The authors estimate transcription rates directly from mRNA decay rates. So if fast translation is associated to mRNA stability, then we'd expect few genes in the bottom left and top right corners of the space. This does not contradict the authors conclusions but the fact that the two axes are not mechanistically independent should be discussed: how is codon optimality distributed in the Crick space? What consequence would have a change in the proportion of co-translational decay versus translation-independent mRNA decay (between genes, between conditions...)?

We now discuss co-translation/decay coupling in the intro and discussion. We note that variation in co-translation stability is smaller than gene-gene variation in transcription rates by about three orders of magnitude: co-translational stability varies by less than one order of magnitude (see e.g. Fig 3F and Fig 7A in Presnyak et al. *Science* 2015), compared to 3 orders of magnitude for transcription. Therefore, axes are not formally independent, but the dependence is small.

On page 38, we now write:

“The quantitative model rests on the assumption that transcription and translation rates can be tuned independently. Although coupling is seen in bacteria [61] and eukaryotes [29], that coupling has itself evolved rather than being an absolute constraint. Studies on synthetic promoters and ribosomal binding sites in *E. coli* indicate that transcription and translation rates can be changed independently over a wide range [41]. In *S. cerevisiae* too, flow-

cytometry analysis of synthetic gene constructs suggest that the eukaryote gene expression machinery can achieve transcription and translation rates that position genes in the depleted region of Crick space (Fig. S2J).

Another coupling relevant to the present study is co-translational stabilization of mRNA [60]. Since rapid translation stabilizes mRNA and mRNA abundance is used to estimate transcription, our estimates of transcription and translation are statistically coupled. However, variation in co-translation stability is smaller than gene-gene variation in transcription rates by about three orders of magnitude: co-translational stability varies by less than one order of magnitude [60], compared to 3 orders of magnitude for transcription. Therefore, our estimates of transcription and translation are not formally independent, but the dependence is small."

3. Another important point is not discussed: the consequence of transcriptional errors. At low transcription/high translation, each RNA 'error' is largely represented among protein molecules; whereas at high transcription/low translation, each error is carried only on a minority of protein molecules. This means that in one case (top left corner of the space) the sources of errors are exposed to function/selection and in the other case (depleted region) they are not. The implication of this should be discussed, especially as error-rates differ between genes. For example, how do the error rates of Gout et al. PMID 29062891 distribute in the Crick space?

We thank the reviewer for this constructive suggestion. We have now added a new figure (Fig. S3F) illustrating how error rates differ between genes. We now discuss transcription errors in the manuscript on pg 39-40:

"We proposed here that a tradeoff between economy and precision in protein abundance explains the depletion of genes with high transcription and low translation. However, the rate of transcription and translation could also impact protein sequence accuracy. For example, the risk that a transcriptional error propagates to many proteins could be reduced by increasing or decreasing transcription, leading to potential tradeoffs between sequence accuracy, precision and economy. At present, tradeoffs involving protein sequencing accuracy are not supported by the data. To see why, we note that changing transcription only changes the probability of error in protein sequence if the error rate depends on the transcription rate. If q is the probability that a newly transcribed mRNA contains an error, the probability that protein synthesis will proceed from an mRNA that carries an error is

$$q \beta_m / (q \beta_m + (1 - q) \beta_m) = q$$

which does not depend on the transcription rate β_m . Thus, changing transcription only changes protein sequence accuracy if the error rate depends on the rate of transcription. Measurements of error rates for thousands of genes in *S. cerevisiae* show no correlation between transcription error rates and transcription rates [27]. Variations in error rates over Crick space are found in a narrow range between 4.0×10^{-6} nucleotide⁻¹ and 6.1×10^{-6} nucleotide⁻¹, with no clear correlation with transcription, translation or the translation / transcription ratio (Fig. S3F). Thus, considerations of protein sequence accuracy are unlikely to deplete genes with high transcription and low translation rate."

4. The population of red points in Fig2E should be presented as a density plot (as for other panels). Not clear where the highest density is and what fraction maps to the depleted region, this is important.

We now produce a density (Fig. S2I). The median bp/bm ratio is much lower for synthetic genes compared to endogenous genes (10^2 vs $10^{3.5}$, $p < 10^{-15}$ at Mann-Whitney's test). 25% of

synthetic genes are in the depleted region vs 1% of endogenous genes.

5. A major value of the study is to accurately predict k values from the trade-off model. For this reason, the test shown in Fig S4D should be better explained: why would anyone expect "a significant correlation between these constants and measured k " as written in legend? What was done here exactly?

We thank the reviewer for this opportunity to clarify this test. We performed this test to challenge the theory. The theory would have little value if any random cellular constant also predicts the depleted region of Crick space. Thus, we verified that the theory can predict the position of the depleted region, while the 10 other cellular constants which we gathered for this project cannot.

The legend of Fig. S4D now reads: "Individual constants cannot predict the position k of the boundary of the depleted region. We correlated the position of the boundary ($\log k$) of the depleted region with 1. the predictions from the theory (Equ. 64) and 2. the 8 cell biology constants listed in Table 1 (number of mRNAs N_m and protein N_p per cell, cell volume V , growth rate μ , active protein decay rate α_{deg} , effective protein decay α_p , mRNA decay α_m , noise floor c_{v0}), plus the maximal translation rate β_{max} and total transcription output $\sum \beta_m$ (Table 1). We tested for significant correlations between these constants (in log) and measured $\log k$ using Pearson's test. The measured $\log k$ correlate significantly with the predictions of the theory. No significant correlation is found between any of the 10 cell biology constants and measured $\log k$."

6. Terms m and $\alpha.m$ appear in main text page 8 without a definition. In methods, $\alpha.m$ is defined as the median decay rate, but in main text I assume that in the expression of C it is gene-specific. This must be clarified.

On page 2, we now define α_m : " α_m [1/h] and α_p [1/h] are the rates of mRNA and protein decay by dilution and degradation (Methods)."

On page 9, we now define m and specify that C is gene specific: " $C = c_m \ln m \alpha_m = c_m \ln m \beta_m / m$ quantifies the cost of transcription per mRNA molecule m for this gene"

7. Regarding the synthetic yeast promoters data of refs 33 & 34: these promoters confer varying levels of transcription rates and noise, with presumably similar translation rate (same coding sequence). As for the synthetic E. coli data, what fraction of them populate the depleted region?

We thank the reviewer for suggesting to reproduce our analysis of synthetic gene constructs in Yeast. We have now used the flow cytometry data of Sharon et al. (*Genome Research* 2014) to estimate transcription and translation rates of 4172 synthetic promoter in yeast. 32% of the synthetic promoters are found in the depleted region of Crick space while only 1% of endogenous genes are in the depleted region. This result confirms the finding we previously made in *E. coli* that 25% of synthetic promoters are found in the depleted region of Crick space. We conclude that the new analysis strengthens the finding that the depleted region of Crick space is accessible to the gene expression machinery.

We now added a new figure (Fig. S2J) and edited the manuscript on pg 6: "In *S. cerevisiae* too, 32% of synthetic promoters from the library of Sharon et al. fall in the depleted region

(Fig. S2J).” Page 73 now reads: “In *S. cerevisiae*, 32% of 4192 synthetic constructs of Sharon et al. fall in the depleted region of Crick space. Transcription and translation were estimated from abundance and noise measurements (26) collected by flow cytometry.”

REVIEWERS' COMMENTS:

Reviewer #1 (Remarks to the Author):

My comments have been adequately addressed.

Reviewer #2 (Remarks to the Author):

I am stratified with the authors' response in general. I will let the editors and the other reviewers make the decision.

Reviewer #3 (Remarks to the Author):

I am satisfied with the revisions made by the authors, expect for one thing. Regarding my comment #3, I thank the authors for producing Fig. S3F which is informative and convincing. However, their additional consideration of error rates along transcription-only changes is irrelevant (what matters here is a joint change of transcription and translation). Thus, the added text on page 39 from "At present, tradeoffs involving protein sequencing..." to "...transcription error rates and transcription rates", including the formula related to q , should be removed.

Reviewer #3 (Remarks to the Author):

I am satisfied with the revisions made by the authors, expect for one thing. Regarding my comment #3, I thank the authors for producing Fig. S3F which is informative and convincing. However, their additional consideration of error rates along transcription-only changes is irrelevant (what matters here is a joint change of transcription and translation). Thus, the added text on page 39 from "At present, tradeoffs involving protein sequencing..." to "...transcription error rates and transcription rates", including the formula related to q , should be removed.

We have now removed the added text on page 13 of the supplementary material as suggested by the reviewer.